# The Relative Influence of Dune Aspect Ratio and Beach Width on Dune Erosion as a function of Storm Duration and Surge Level

Michael Itzkin[1], Laura J. Moore[1], Peter Ruggiero[2], Sally D. Hacker[3], and Reuben G. Biel[1]

[1]Department of Geological Sciences, University of North Carolina, 104 South Road, Mitchell

Hall, Campus Box 3315, Chapel Hill, North Carolina 27515, USA

[2]College of Earth, Ocean, and Atmospheric Sciences, Oregon State University, 104 CEOAS Administration Building,

Corvallis, OR 97331, USA

[3]Department of Integrative Biology, Oregon State University, 3029 Cordley Hall, Corvallis, OR 97331, USA

*Correspondence to:* Michael Itzkin (mitzkin@unc.edu)

**Abstract.** Dune height is an important predictor of impact during a storm event given that taller dunes have a lower likelihood of being overtopped than shorter dunes. However, the temporal dominance of the wave collision regime, wherein volume loss (erosion) from the dune occurs through dune retreat without overtopping, suggests that dune width must also be considered when evaluating the vulnerability of dunes to erosion. We use XBeach, a numerical model that simulates hydrodynamic processes, sediment transport, and morphologic change, to analyse storm-induced dune erosion as a function of dune aspect ratio (i.e., dune height versus dune width) for storms of varying intensity and duration. We find that low aspect ratio (low and wide) dunes lose less volume than high aspect ratio (tall and narrow) dunes during longer and more intense storms when the beach width is controlled for. In managed dune scenarios, where sand fences are used to construct a "fenced" dune seaward of the existing "natural" dune, we find that fenced dunes effectively prevent the natural dune behind them from experiencing any volume loss until the fenced dune is sufficiently eroded, reducing the magnitude of erosion of the natural dune by up to 50%. We then control for dune morphology to assess volume loss as a function of beach width and confirm that beach width exerts a significant influence on dune erosion; a wide beach offers the greatest protection from erosion in all circumstances while the width of the dune determines how long the dune will last under persistent scarping. These findings suggest that efforts to maintain a wide beach may be effective at protecting coastal communities from dune loss. However, a trade-off may exist in maintaining wide beaches and dunes in that the protection offered in the short-term must be considered

in concert with potentially long-term detrimental effects of limiting overwash, a process which is critical to maintaining island elevation as sea level rises.

## 1. Introduction

Foredunes provide the first line of defense for coastal communities against overwash and inundation during storms. To this end, a tall dune is often considered ideal for mitigating storm impacts as its height is less likely to be exceeded by the storm-induced total water level (TWL) (Biel et al., 2017; Sallenger, 2000; Seabloom et al., 2013; Stockdon et al., 2006). Considering that wave runup is more likely to impact the dune face than to exceed the crest, collision (Sallenger, 2000) is the most temporally common impact regime during a storm (Brodie et al., 2019; Stockdon et al., 2007). Considering that collision leads to scarping, the width of the dune is an important predictor of how vulnerable a dune is to erosion during a storm (i.e., Leaman et al., 2020). While dune height change is an important metric to consider when measuring storm impacts (e.g., Long et al., 2014), other measures of dune erosion, such as volume loss (e.g., Durán et al., 2016; Larson et al., 2004), may better describe the overall change in morphology by accounting for changes in height as well as width (Figure 1). While the role that dune height plays in determining storm impact has been well studied, it is less clear how the shape (aspect ratio) of the dune influences the style and magnitude of erosion. The dune aspect ratio metric, which we analyse in this paper, allows us to quantify how dunes erode under persistent scarping conditions (related to the width of the dune) while still considering the susceptibility of the dune to overwash (related to the height of the dune).

Dunes form as a result of biophysical feedbacks between aeolian sediment transport and vegetation growth (Durán and Moore, 2013; Hesp, 2002; Houser et al., 2015; Maun, 1998; Stallins and Parker, 2003). Dune cross-shore position and dune height are controlled by the distance between the shoreline and the seaward limit of vegetation, with longer distances typically being associated with the formation of taller dunes (Durán and Moore, 2013; Hesp, 2002; complexities occur on prograding shorelines, see Moore et al., 2016) While vegetation zonation controls the positioning and height of dunes, the dominant plant species can influence overall dune shape (e.g., Biel et al., 2019; Hacker et al., 2012; Woodhouse et al., 1977; Zarnetske et al., 2010, 2012). Dune grasses that tend to grow more horizontally than vertically will tend to form dunes that are shorter and wider, and vice-versa. For example, Hacker et al. (2012) found that dunes in the Pacific Northwest (PNW) formed in the presence of *Ammophila arenaria* were typically taller and narrower than dunes formed in the presence of *Ammophila breviligulata*. *Ammophila arenaria* grows more vertically while *A. breviligulata* grows more laterally and their respective dune morphologies reflect that

difference. As a result, coastal foredunes in the PNW where *A. breviligulata* is dominant may be exposed to a greater risk of overtopping (Seabloom et al., 2013). In a study of east coast dune grass species (*A. breviligula* and *Uniola paniculata*), Woodhouse (1977) found that dunes formed in the presence of *A. breviligulata* were shorter and wider than those formed in the presence of *U. paniculata* under similar environmental conditions; a result confirmed by the recent work of Hacker et al. (2019) and Jay et al. (in revision). Changing environmental conditions may also drive shifts in the range of dominant dune grass species, for example Goldstein et al. (2018) found that there is a northern shift in the range of *U. paniculata* along the east coast of the United States consistent with latitudinal temperature trends. This may cause a concomitant shift in dune shape along the northeastern United States to a taller but narrower morphology (i.e., higher aspect ratio). Given this, and a similar change in dune morphology in the PNW driven by changes in dune grass species, it is important to understand how such changes in dune shape will impact coastal erosion given that taller/narrower dunes may be more protected from overwash but more likely to be eroded via scarping.

In addition to natural controls on dune growth and dune shape, human modifications to beach and dune systems may involve constructing a new foredune, or fortifying the existing foredune to make it more resistant to erosion (e.g., Elko et al., 2016; Nordstrom et al., 2000; Nordstrom and Jackson, 2013), often through the use of sand fencing (e.g., Anthony et al., 2007; Charbonneau and Wnek, 2016; Jackson and Nordstrom, 2018; Miller et al., 2001). In a study on the geomorphic effects of sand fencing, Itzkin et al. (2020) demonstrated that foredunes are lower in elevation in areas where sand fences are constructed but that the dune system overall is substantially wider than foredunes in areas without sand fences. Other management actions, such as allowing vehicles to drive on beaches and beach raking, can also generate trade-offs between dune growth and coastal protection (Defeo et al., 2009). In a study on the effects of dune grass removal to restore Western snowy plover habitats in the PNW, Biel et al (2017) used XBeach (Roelvink et al., 2009) to explore dune erosion and found that where beachgrasses were removed, dunes maintained a lower elevation and were predicted to be more vulnerable to erosion compared to foredunes where beachgrasses were not removed and grew to a stable elevation. Beach nourishment may also be used to widen the beach (and decrease its slope), limiting wave impacts to the dune and stimulating dune growth (e.g., Cohn et al., 2019; Van Puijenbroek et al., 2017; Ruggiero et al., 2001, 2004).

The main goal of the work presented here is to assess how dunes erode during a single storm as a function of dune aspect ratio, beach width, and sand fence construction. While previous studies of storm impacts have primarily

focused on dune height (e.g., Long et al., 2014; Sallenger, 2000; Stockdon et al., 2007), recent studies suggest that dune width may also be a key predictor for how much dune erosion will be experienced (Leaman et al., 2020). Additionally, although beach width has been posited as a strong predictor of dune erosion (e.g., Burroughs and Tebbens, 2008; Claudino-Sales et al., 2008; Itzkin et al., 2020; Silva et al., 2018), we seek to quantify and understand the relative role of beach width in dune erosion processes. To achieve these goals, we used XBeach (Roelvink et al., 2009), a process-based numerical model that simulates nearshore hydrodynamics, sediment transport, and morphologic change over storm time scales and has been well validated for simulating dune response to storms (e.g., Cohn et al., 2019; McCall et al., 2010; Vousdoukas, Ferreira, Almeida, & Pacheco, 2012). We address the following specific questions: (1) How does storm duration affect volumetric dune erosion as a function of foredune aspect ratio? (2) How do variations in storm TWL affect volumetric dune erosion as a function of foredune aspect ratio? (3) How does the morphology of the beach (i.e., width and slope) affect volumetric dune erosion independent of foredune aspect ratio. Finally, we also compare our model results with observed pre- and post-storm lidar and field profiles from the North Carolina coast to ground-truth our numerical analyses.

## 2. Methods

### 2.1 Beach and Foredune Morphometrics

To track changes in dune and beach morphology throughout our simulations, the following morphometrics are calculated at every model time step: dune aspect ratio, dune volume, overwash volume, beach width, dune toe erosion, and wave energy reaching the toe of the dune over time as the dune evolves. The dune aspect ratio was calculated as the height of the natural dune from $D_{high}$ to $D_{low}$ divided by the width of the dune from $D_{heel}$ to $D_{low}$ (Figure 1). The dune volume is calculated by integrating over the portion of the profile contained within the original cross-shore location of the dune ($D_{low}$ to $D_{heel}$) in the first-time step and above the $D_{low}$ elevation (0.6 m, NAVD88). Beach width is calculated as the cross-shore distance between MHW and $D_{low}$ at every time step. Given that pre-storm $D_{low}$ was held constant across all simulations, the beach slope (measured between MHW and $D_{low}$) is inversely proportional to the beach width in our simulations (i.e., beach slope decreases as beach width increases). Dune toe retreat was measured as the final minus initial cross-shore position of the dune toe. Wave energy is counted as the cumulative amount of wave energy at the dune toe as it evolves throughout the simulation.

### 2.2 Observed Foredune Profiles

We use changes between in situ beach profiles from 2017-2018, which capture the influence of Hurricane Florence, to test model results. Topographic profiles were measured along Bogue Banks (BB), NC, USA using a Trimble Real Time Kinematic Global Positioning System (RTK-GPS) in October of 2017 and 2018. Twenty-two total profiles were surveyed with an alongshore spacing of 1-2 km. For each field transect, we calculate the aspect ratio,

beach width, and dune volume using the methods described above. We also use BB as a reference location for developing model inputs and synthetic dune profiles. BB is a roughly 40 km-long developed barrier island in the Outer Banks of North Carolina along which the dune system has been modified through the use of sand fencing. This management action has led to the development of a relatively low yet wide dune where fences are present. We use a profile from Fort Macon, a non-fenced portion of the island, as a reference beach and dune profile to generate our

synthetic dune shapes as described below.

**2.3 Synthetic Dune Profiles**

We created a set of synthetic beach-dune profiles using a LiDAR-derived initial reference profile from Fort Macon, North Carolina, USA (Figure 2, 3). We fit an exponential curve (Komar and McDougal, 1994) to a measured bathymetric profile to extend the beach profile to the buoy depth of 30.5 m. The height of the reference dune was

125 increased (decreased) in 20% intervals as:

$$H_f = H_r * (1 - \frac{stretch}{100})$$ (1)

where $H_f$ is the post-stretch dune height, $H_r$ is the height of the reference dune, and stretch is a multiple of 20 between -60 and 60. Every increase (decrease) in dune height is paired with a decrease (increase) in dune width such that:

$$W_f = \frac{W_r}{1 - \frac{stretch}{100}}$$ (2)

where $W_f$ is the post-stretch dune width, $W_r$ is the reference dune width. This method of simultaneously modifying dune height and width allowed for dune volume to be conserved (and therefore essentially held constant across simulations) and resulted in a suite of beach profiles with aspect ratios (i.e., dune height divided by dune width) ranging from 0.02 to 0.27 and dune volumes between 50.6 and 53.5 $m^3$/m. While we could have controlled for other components of the dune shape (i.e., steepness, height, width), our analysis primarily focuses on volume loss and so

our approach to creating synthetic dune shapes allows all simulations (aspect ratios) to have nearly the same starting volume. Additionally, because dune aspect ratio changes as dune height, width, and/or steepness change, characterizing dune shape using the aspect ratio allows us to capture modifications to dune morphology. For completeness, we describe the initial dune shape parameters in Table 1. Though beyond the scope of this study,

exploring alternative methods of modifying the dune shape (e.g., maintaining the dune face slope but increasing the width) would be useful in better understanding how individual components of the dune shape affect erosion during a storm.

We adjusted the position of the synthetic "natural" dune on the profile to create four different cross-shore configurations: 1) dune toe ($D_{low}$) positions aligned, 2) dune crest ($D_{high}$) positions aligned, 3) dune heel ($D_{heel}$) positions aligned, and 4) dune toe ($D_{low}$) positions aligned with a sand-fenced dune seaward of the synthetic natural dune (Figure 3A-D). Simulations with the $D_{low}$ position aligned control for the morphology of the beach fronting the dune to isolate the role of dune morphology on erosion. Simulations with the $D_{high}$ positions aligned may be more representative of a natural setting in which wider beaches are backed by taller dunes. Simulations with $D_{heel}$ positions aligned may be representative of a managed shoreline, where the dunes are widened seaward and thus share a common heel position regardless of dune height. In both the $D_{high}$ and $D_{heel}$ aligned scenarios, the beach width increases proportionally with the aspect ratio of the dune. The fourth configuration (Figure 3D) represents a dune complex that arises when sand fences are placed on managed shorelines. The series of fenced profiles is the same as the series synthetic natural dune profiles in which the $D_{low}$ position is aligned (Figure 3A) except that we added a gaussian curve on the seaward side of the dune to represent the presence of a typical fenced dune shape (Itzkin et al., 2020). Similar to the simulations with $D_{low}$ aligned, our fenced dune simulations control for the morphology of the beach fronting the fenced-natural dune system (Figure 3D) and therefore isolate the role of the fenced dune in mitigating natural dune erosion.

To demonstrate that the synthetic dunes are consistent with observed morphologies, and that it is reasonable to hold the volume constant for a dune while modifying its aspect ratio, we compared the aspect ratios and volumes of the synthetic dune profiles with those of LiDAR-derived dune profiles from Bogue Banks measured between 1997 and 2016 (Figure 4). The aspect ratios of dunes on Bogue Banks range from close to zero up to approximately 1.08. About 90% of all profiles fall within the range of aspect ratios of our synthetic profiles. While the lidar data is extracted in locations where dunes are present, the lowest aspect ratios explored in this study are essentially flat, representing conditions in which a dune is absent. Dune volumes on Bogue Banks range up to 350 m$^3$/m with the modelled value representing the 80[th] percentile. Given the relatively weak relationship between dune volume and dune aspect ratio (with the aspect ratios used in this study having a wide range of associated volumes; Figure 4), maintaining a relatively constant dune volume while varying the dune aspect ratio in our model simulations is reasonable.

**2.4 Synthetic Storm Hydrographs**

We created a set of synthetic storms for use in the model simulations by using Hurricane Matthew as a reference storm and then increasing its duration by up to 48 hours (Figure 5). Hurricane Matthew moved northward along the North Carolina coast on the afternoon of October 8, 2016, generating approximately 1 m of storm surge and significant wave heights (Hs) of approximately 7.5 m. To capture the full spin up, peak, and relaxation of the storm, we used wave (Hs, peak period, Direction) and tide data for October 7-10, 2016, from the nearest NOAA National Data Buoy Center (NDBC) buoy (41159; Onslow Bay, NC; depth = 30.5 m) and NOAA tide gauge (Station CLKN7; Beaufort, NC). We used linear wave theory to transform the wave parameters to the 30.5 m depth contour to account for shoaling and refraction with the transformed wave data used as input for XBeach.

To represent a longer duration storm than the base storm, we used the Hurricane Matthew storm time series to identify a 12-hour window centered on the timing of peak storm surge. We then interpolated all hydrodynamics (i.e., Hs, peak period, direction, and still water level) within this temporal window onto a +12hr, +18hr, +24hr, +36hr, and +48hr temporal "grid," effectively increasing the storm's duration by up to two days. We held constant the spin up (rising hydrograph) and relaxation (falling hydrograph) of the storm for all simulations. For all storm durations, we created a version in which the surge is unmodified (1.0x), decreased by 50% (0.5x), and increased by 50% (1.5x). In total, this yielded 18 different synthetic storms (Figure 5).

The duration of our synthetic storms varied from 73 hr to 122 hr, and the surge in our synthetic storms varied from ~0.5 m to ~1.25 m. These values are comparable to other storms that have recently affected the North Carolina coast, including Tropical Storm Joaquin (duration ~144 hours) and Hurricane Florence (Duration ~48 hours) (Figure 6). Water levels during Tropical Storm Joaquin were elevated for 6 days, which is comparable to the total duration of the +48 hr storm time series. Peak surge during Hurricane Florence was ~1.6 m, a similar order of magnitude to the maximum storm surge we used in the synthetic time series.

**2.5 Foredune Erosion Simulations**

We used the XBeach (Roelvink et al., 2009) model to simulate the effects of the synthetic storms described in Section 2.4 on the profiles described in Section 2.3. We ran XBeach (version 1.23.5465) in 1D-hydrostatic mode with the break parameter set to roelvink_daly and the gamma parameter set to 0.52 to better capture the effect of swash processes on the reflective beach profiles (Roelvink et al., 2018). We also adjusted parameters related to wave breaking and dry sediment transport in order to more realistically simulate dune erosion processes given XBeach's

tendency to overestimate erosion with default settings (Palmsten and Holman, 2011, 2012; Palmsten and Splinter, 2016; Splinter and Palmsten, 2012). XBeach erodes the profile by comparing the slopes to the dryslp (if a cell is dry) parameter or wetslp (if a cell is wet) to determine how much erosion should occur to maintain these values. Palmsten and Holman (2011, 2012) showed that wet sand can sustain much steeper scarps. By using a particularly high value for the dry slope (*dryslp = 4*) we allow the dunes to maintain much steeper, and more realistic, scarps during storms

(Palmsten and Splinter, 2016). This allows us to better understand how the dune is eroding under collision when the dune is actively scarping by comparing dune toe migration to dune volume loss. A full list of non-default parameters can be found in Table 2. We gridded the profiles described in Section 2.3 for use with XBeach; subaerial spacing is 1 m and subaqueous spacing varies from 5-20 m to decrease computational cost.

We grouped the simulations into 12 "experiments" that encompass all combinations of dune configuration

(i.e., toe-aligned, crest-aligned, heel-aligned, and fenced) and storm surge modification (i.e., 0.5x, 1.0x, 1.5x) (Figure 7). Within each experiment, we simulated all combinations of dune aspect ratios and storm durations, which resulted in a total of 504 simulations (12 experiments with 42 simulations per experiment). We note that all dune erosion during the simulations occurred in the collision regime (Sallenger, 2000), unless stated otherwise. Further, fenced profiles do not contain any structural reinforcement arising from the presence of a fence that might otherwise limit how the dunes

are eroded; although the fenced dune itself limits erosion of the natural dune behind it, the authors are unaware of any studies showing an effect on dune erosion during a storm from the fence itself.

## 3. Modelling Results

### 3.1 Erosion on Synthetic Dunes

Overall, our simulations for dunes without fences show that losses in foredune volume are greater with higher

storm surges, longer storm durations, or when dunes are located closer to the shoreline (represented by the dune toe-aligned scenarios; Figure 8). Foredunes erode under most simulated conditions (Figures 8, 9, 10, and 12), except when they are situated farther from the shoreline (Figure 11). Additionally, the four different dune configurations included in our analysis (i.e., toe-aligned, crest-aligned, heel-aligned, and fenced) allow us to isolate the amount of erosion attributable to the dune morphology (aspect ratio) versus the amount of erosion attributable to the beach morphology

(width and slope) by establishing a baseline level of dune erosion in the toe-aligned simulations before introducing variations in the beach width in the crest- and heel-aligned simulations and management interventions in the fenced simulations. Below are the specific results of the erosion simulations using the four types of synthetic dune

configurations. We note that while the specific quantitative results relating to dune erosion and wave energy are a function of model setup (e.g., calibration parameters), the setup is applied uniformly to all simulations such that, as we have seen in previous sets of simulations with different parameterizations not presented in this manuscript, we expect the resulting trends to be consistent regardless of how the dune shapes are formulated and how the model parameters are set.

### 3.1.1 Dune Toes Aligned – Isolating the effect of dune aspect ratio on dune erosion

The profiles for all toe-aligned simulations have the same beach width (and slope) (Table 1). Because the only difference between these simulations is the dune morphology, we use them to isolate the effect of dune aspect ratio on dune erosion. Simulations with the dune toes aligned show that there is greater erosion for the high-aspect ratio dunes compared to the low-aspect ratio dunes (Figure 8A, B, C). The increased erosion is clear during low intensity storms where the high aspect ratio dunes lost 19% (~10m$^3$/m) more sediment than the low aspect ratio dunes. For the longest and most intense storms, the difference in volume loss between the high and low-aspect ratio dunes is ≤10 m$^3$/m. As expected, increasing the duration of the storms leads to an increase in the amount of overall erosion, especially for high-aspect ratio dunes. While none of the dunes are completely inundated in our simulations, the dunes (all aspect ratios) lose a significant amount of sediment (>30m$^3$/m, >60%) during the long duration storms.

Although the tall/narrow dunes lose more sediment than the low/wide dunes, the dune toe experiences less retreat regardless of the storm scenario. While the dune toe for the low/wide dune retreats up to 10m during the longest and most intense storms (Figure 8F), the dune toe for the tall/narrow dunes never retreats more than ~5m (Figure 8D). In some cases, the tall/narrow dune toes prograde seaward likely via avalanching by up to ~12m (Figure 8F). For a given storm duration and intensity, the dunes of different aspect ratios are impacted by a comparable amount of wave energy (Figure 8G, H, I). Therefore, since the beach morphology is the same for each of the toe aligned simulations the style of erosion is purely being regulated by the morphology of the dune. High aspect ratio dunes are closer to the angle of repose than low aspect ratio dunes so they tend towards avalanching with sediment piling up at the dune toe. While the low aspect ratio dunes lose less volume than the high aspect ratio dunes, the sediment tends to be lost offshore during the storm. For example, during a relatively short (+10 hours) storm with low surge, dunes with an aspect ratio of 0.1 and 0.2 both lose ~5m$^3$/m of sediment (Figure 8A) and are impacted by ~500Nm/m$^2$ of wave energy (Figure 8G) during the storm. However, despite these similarities in volume loss and wave energy, the sediment for

the lower aspect ratio dune is lost offshore while the sediment from the higher aspect ratio dune is deposited at the toe of the dune via scarping.

**3.1.2 Dune Crests Aligned and dune heels aligned – Isolating the effect of beach width**

Crest- and heel-aligned simulations use the same dune morphologies as the toe-aligned simulations, however, the beach width increases proportionally for the crest- and heel-aligned profiles such that higher aspect ratio dunes are

fronted by wider beaches than the lower aspect ratio dunes. Given that wave runup and erosion during a storm is lower for wider, more gently sloping beaches (i.e., Ruggiero, Holman, & Beach, 2004; Stockdon et al., 2006; Straub et al., 2020), we analyze the effect of beach width on dune erosion (separate from effects of dune aspect ratio; Figure 8). For the simulations in which the dune toe is aligned, beach width is constant for all aspect ratios and thus does not affect dune erosion and retreat. However, because the crest- and heel-aligned dunes can vary in their beach morphology

depending on aspect ratio, this difference leads to wider beaches and might explain decreased erosion for high aspect ratio dunes (Figures 9 and 10). To isolate the effect of beach width, we subtract the amount of dune erosion (i.e., volume, toe position change, and wave energy) that occurred in the toe-aligned simulations (which control for beach width) from the amount of erosion in the crest- and heel-aligned simulations. This calculation yields a positive number for volume change and dune toe migration, representing the volume of sediment preserved in the dune as a

consequence of increasing the beach width, and a negative value for wave energy representing additional wave dissipation provided by the beach for both the crest- and heel-aligned simulations. We find that the widest beaches (associated with the highest aspect ratio dunes) prevent more erosion than the narrowest beaches (associated with the lowest aspect ratio dunes) and that the protection offered by the increased beach width becomes more pronounced as storm duration increases (Figure 9A, B, C and Figure 10A, B, C). Considering all the simulations together (Figure

11), we observe a proportional relationship between the pre-storm beach width and dune volume loss (relative to the equivalent toe-aligned simulation). We also find that the erosion mitigated by wider beaches is even greater under longer and stronger storms (Figure 11).

The final dune toe position is consistently farther seaward of the initial dune toe position for all dunes fronted by wider beaches than it is for the equivalent toe-aligned simulations and this effect was proportional to the beach

width. For example, the toe of the highest aspect ratio dune (toe-aligned) did not migrate during the longest storm with 1.5X surge while the toe of the lowest aspect ratio dune retreated by ~10m during the same storm during the toe-aligned simulations (Figure 8F). During the crest- (heel-) aligned simulations when these same dunes are fronted by

wider (widest) beaches, the post-storm dune toe location of the high aspect ratio dune is located 15m (30m) farther landward compared to those observed in the toe-aligned simulations (where the toe of the high aspect ratio dune does

not change) while the post-storm dune toe of the low aspect ratio dune is unchanged relative to the toe-aligned simulation (Figure 9F, Figure 10F). This change is likely driven through avalanching from the dune as well as wave driven sediment transport to the beach/dune (e.g., Cohn et al., 2019). Wave energy reaching the dune is reduced by up to 6000 Nm/m$^2$ during the most intense storms in simulations with the widest beaches compared to a reduction of 1000 Nm/m$^2$ when the dunes are fronted by a narrower beach (heel-aligned; Figure 10I).

**3.2 Erosion of Synthetic Dunes with Sand Fences**

We performed a suite of simulations using the same dune profiles as the dune toe-aligned scenarios but with a portion of the beach replaced by a fenced dune (Figure 1D). By comparing the results from these simulations with those from the toe-aligned simulations (Figure 11) we quantify the effectiveness of artificial dunes (formed via the emplacement of sand fences) under varying storm scenarios while controlling for the effects of beach width. We find

that the fenced dunes prevent more volume loss (up to ~20 m$^3$/m) as the surge increases (Figure 12A, B, C) however, for any given surge level, there is a minimal (<10 m$^3$/m) difference in the amount of dune toe retreat mitigated by the fenced dune between the longest and shortest storms (Figure 12D, E, F). Additionally, the aspect ratio of the dune behind the fence plays a minimal role in influencing volume loss (because the natural dune is never impacted until the fenced dune is eroded) except in the case of the most intense storms when the lowest aspect ratio dunes performs

better than the higher aspect ratio dunes (Figure 12C). Finally, the fenced dunes reduced the amount of wave energy impacting the non-fenced dune by up to ~2500 nm/m$^2$ during the most intense storms (Figure 12I). Unlike their influence on volume loss and dune toe migration, the influence of fenced dunes on wave energy dissipation demonstrates a relationship with storm duration wherein more energy is dissipated during longer storms (Figure 12G, H, I). This result is expected given that the wave energy is related to impact hours and longer storms lead to the dune

being impacted longer.

**3.3 Comparisons with Field Surveys**

We compare our model results to the observational field surveys conducted along Bogue Banks, NC, before (2017) and after (2018) Hurricane Florence. The field data show a weak relationship between dune aspect ratio and erosion (sand volume loss). Similar to model results for the toe-aligned (constant beach width) dune configurations,

comparing volume loss for dunes fronted by equally wide beaches, profiles with a lower aspect ratio dune generally

experienced similar or even less erosion than high aspect ratio dunes with the same beach width. For example, the field profiles with a beach width of ~30 m show slightly more erosion for the higher aspect ratio dunes compared to the lower aspect ratio dunes (Figure 13).A strong trend occurs with respect to the beach width, in which erosion significantly decreases with increasing beach width, regardless of aspect ratio (Figure 13). Foredunes with beach widths greater than 40 m all experience dune growth between 2017-2018 despite the effects of Hurricane Florence during this period (Figure 13). Although the model does not simulate aeolian transport induced dune growth, we note that in simulations with low to moderate surge (0.5x-1.0x), sand volume loss decreases to zero or near-zero for beach widths greater than 40m (Figure 9).

## 4. Discussion

### 4.1 Effects of Aspect Ratio on Dune Erosion

The storm impact scale for barrier islands described by Sallenger (2000) relates the elevation of the dune crest and dune toe to the TWL as a means of categorizing impacts within four possible impact regimes: swash, collision, overwash, and inundation. Each storm impact regime has a corresponding mode of dune erosion associated with it, ranging from none (swash regime) to potentially complete loss of the dune (inundation regime). A key implication of this storm impact scale is that a taller dune should provide better protection from storms because it is less likely to be overtopped. Previous studies (e.g., Brodie et al., 2019; Stockdon et al., 2007) have suggested that collision is the most common, but temporary, storm impact regime to impact foredunes. Collision occurs when the TWL impacts the face of the dune, causing scarping and dune retreat. This suggests that a dune which is more resistant to the effects of collision may offer the greatest degree of protection so long as it is not overtopped, especially given that the dune will likely be experiencing collision for a longer time under longer duration storm events. To this end, while the role of dune width in mitigating hazard exposure has been explored qualitatively (Davidson et al., 2020; Leaman et al., 2020), detailed quantitative assessments were not available prior to the present study. Further, the relative role of dune morphology (height and width) versus the beach width in limiting exposure to coastal hazards has been identified as a necessary avenue for future research (Davidson et al., 2020), an avenue our results shed light on.

When controlling for the beach width (i.e., the dune toe-aligned scenario), we find that the lower aspect ratio dunes eroded less than the higher aspect ratio dunes when both are in the collision regime. High aspect ratio dunes are more likely to collapse when scarped because avalanching is likely to occur as the dune face slope approaches the

angle of repose. Avalanching also likely explains accretion occurring at the dune toe for the high aspect ratio dunes (Palmsten and Splinter, 2016; which may also be accompanied by dune crest retreat as the dune face is eroded beyond the initial crest position). Although high aspect ratio dunes are better equipped to prevent overwash, there is the potential for them to be completely eroded during the longest storms due to persistent scarping, undercutting, and avalanching. In contrast, low aspect ratio dunes can withstand long duration storms if storm surge is not sufficiently high to cause overwash. While the height of the dune may be an appropriate predictor for overwash, the overall aspect ratio may better describe how dunes will erode with respect to storm duration and intensity. While the most resistant dune shape would be a dune tall enough to minimize the likelihood of overwash and wide enough to prevent significant loss of sediment and dune toe retreat via scarping, when resource limitations (such as sediment availability) require prioritizing management interventions our results suggest opting for widening dunes rather than building them vertically, may be worth considering.

While dune morphology plays a primary role in describing how dunes erode, particularly with respect to whether sediment is piled at the toe of the dune (high aspect ratio dunes) or transported offshore (low aspect ratio dunes), it plays a secondary role to the beach morphology in terms of explaining the amount of erosion that will occur. Across simulations in which the beach width varied (i.e., the crest-aligned and heel-aligned scenarios), there is a clear relationship between the amount of erosion prevented (relative to the toe-aligned scenarios) and the width of the beach (Figure 10), Wider beaches lead to less sediment loss from the dune and a more seaward post-storm location of the dune toe. During the longest and most intense storms, the widest beaches lead to a dissipation of up to ~66% of the wave energy reaching the dune (Figure 9I), corresponding to the simulations with the least amount of observed erosion. This result, combined with the results of the toe-aligned scenarios confirms that beach width is the primary control on the volume of sediment eroded from dunes during storms. Our results further suggest that dune width and dune height also affect the volume of sediment eroded by determining the "longevity" of the dune (thereby affecting the transition from avalanching to overwashing, e.g.) under erosive conditions.

**4.2 Sand Fences and Beach Nourishment on Foredune Erosion**

We assess the effects of sand fencing on the mitigation of dune erosion by comparing the results of the fenced simulations with the toe-aligned simulations. We find that the small dune formed by fencing can significantly decrease dune erosion by providing a barrier that must be removed by erosion before the "natural" dune behind it is impacted. In our simulations, the fenced dune was not sufficiently eroded until ~60 hours into the storm, which prevented the

dune behind it from experiencing the peak of the storm (Figure 4). The key dynamic in this case (regardless of actual storm duration), was that the fenced dune was sufficiently high and wide to protect the natural dune until the peak of the storm had passed. Thus, the aspect ratio of the natural dune is secondary to the morphology of the fenced dune in

providing protection to back-barrier environments (a taller and/or wider fenced dune would offer even greater protection). Charbonneau and Wnek (2016) demonstrated that fenced dunes (especially when paired with a nourishment) can form quickly (on the order of months) meaning that not only can the fenced dunes effectively prevent storm-induced erosion, but it is possible for them to recover prior to the next storm if the frequency of storm impacts is sufficiently low, and assuming the fences are still present following the storm or are re-built. Additionally, dunes

built by sand fences may be paired with vegetation planting to assist in sand trapping efficiency and dune stabilization (Bossard and Nicolae Lerma, 2020; Nordstrom and Jackson, 2013). While we did not simulate management interventions that would widen the dune (e.g., dune grass planting, dune scraping, sand ramps), the effect of these interventions could be hypothesized from our results. Any management strategy that widens the dune but does not add to its elevation will cause the dune to assume a lower aspect ratio than it had in its pre-management state. Further, if

management is not paired with a beach nourishment (i.e., Itzkin et al., 2020) then the wider dune will likely come at the cost of a slightly narrower beach. The lower aspect ratio (Figure 8) could serve to reduce erosion, but this potential decrease in erosion may well be offset by increased erosion associated with the narrower beach (Figure 10).

We also effectively considered the role of beach nourishment by varying beach width for the dune crest-aligned and dune heel-aligned scenarios. For a given dune aspect ratio, the only difference between the toe-aligned,

crest-aligned, and heel-aligned simulations was the beach width. Beach nourishment (i.e., widening of the beach) appears to have a greater impact on preventing dune erosion than any management action that could be taken to alter the aspect ratio of the dune. XBeach simulations with the dune crest- and heel-aligned have a variable beach width that is proportional to the dune aspect ratio. For a given dune aspect ratio and wave duration and intensity, the only difference between the simulations is the increase in beach width (toe-aligned < heel-aligned). This increase in beach

width decreases beach slope ($\beta_f$), which lowers incident band swash (e.g., Ruggiero et al., 2004) and total wave runup (Stockdon et al., 2006), reducing the likelihood of dune erosion. We see this effect in our results, which show up to a 100% reduction in dune erosion between the toe- and heel-aligned simulations (Figure 8). The strong inverse relationship between beach width and dune erosion (Figures 10 and 11) suggests that regardless of the aspect ratio of a foredune, widening the beach can be sufficient for preventing overwash during most storms and will be a more

effective strategy for increasing coastal protection than re-building the dune or installing sand fences; although pairing

sand fences with a wide beach via nourishment would offer the greatest overall reduction in natural dune volume loss.

It is important to recognize that although management initiatives such as widening beaches and building

dunes with particular aspect ratios can be effective at mitigating erosion during a single storm, these actions may have

effects that are undesirable in the long-term as the effects of multiple storms compound. For example, overwash

facilitates barrier rollover—a process that is necessary if islands are to maintain subaerial exposure in the long-term

future as sea level rises (e.g., Leatherman, 1979; Moore et al., 2010; Lorenzo-Trueba & Ashton, 2014; Rogers et al.,

2015). Thus, constructing dune and beach systems that reduce the amount of overwash that would otherwise naturally

occur may inhibit rollover, thereby increasing the likelihood of eventual barrier drowning (e.g., Magliocca et al., 2011;

Rogers et al., 2015).  This is a concern even in the presence of expected ongoing beach nourishment both because

overwash-induced increases in island interior, and back-barrier elevation are necessary to prevent drowning from the

backside and because it is not feasible to continue beach nourishment indefinitely along all developed barriers.

**5. Conclusions**

In this study we used XBeach to analyze how coastal foredunes erode during a storm as a function of their

aspect ratio (height and width), beach width, and the presence of various management interventions (i.e., sand fencing

and beach nourishment). We find that low aspect ratio (lower and wider) dunes are more resistant to erosion from

increased storm duration than high aspect ratio (taller and narrower) dunes, although high aspect ratio dunes offer

greater protection against more intense storms. For low aspect ratio dunes, eroded sediment is lost offshore whereas

the high aspect ratio dunes lose greater amounts of sediment through persistent scarping, more of the sediment is

preserved at the toe of the dune, consistent with the piling up of sediment at the base of the dune as would occur as a

result of avalanching. In addition, dunes built by sand fences reduce the amount of erosion experienced by the foredune

behind the fenced dune because they create a barrier that must first be eroded through or overtopped. Although

modifying the dune aspect ratio and/or constructing a fenced dune does alter the amount of erosion experienced as

storm characteristics vary, we find that the greatest protective service in all instances is offered by a wide beach; a

finding that is consistent with previous assertions and also supported by our limited observations of dune erosion in

the field.

Our results indicate that a tall, wide foredune fronted by a fenced dune and a wide beach offer the greatest

protection from erosion. Given the challenges of achieving such a foredune morphology in the face of rising sea level

and within resource limitations (i.e., sand availability, cost, etc.), our findings suggest that the greatest increase in short-term protective service can be achieved by widening beaches, regardless of the frontal dune morphology.

However, although our work could be used to demonstrate that widening beaches through nourishment may provide more protection from flooding in the near-term than artificially constructed dunes, we note that this is costly, may be infeasible, and is irresponsible over the long-term given that these management initiatives reduce overwash flux, which is essential for barrier islands to maintain elevation as sea level continues to rise (i.e., Lorenzo-Trueba & Ashton, 2014; Magliocca et al., 2011; Rogers et al., 2015). Alternative strategies for widening beaches would also have a

similar protective effect (e.g., managed retreat; Cutler et al., 2020; Gibbs, 2016) while allowing for more of the natural processes to occur that allow an island to evolve and persist in the face of rising sea levels. Ultimately, pairing wide dunes and wide beaches to provide protection from the most common storm scenarios may balance the need for protection from most storms, but this must be considered against a longer-term need for the island to rollover and maintain a subaerial surface via overwash processes lest it risk drowning.

**6. Data Availability**

The data analyzed in this submission are available at Zenodo: https://doi.org/10.5281/zenodo.4744544. LiDAR data is available from NOAA: https://chs.coast.noaa.gov/htdata/lidar2_z/geoid18/data/5184/.

**7. Author Contribution**

MI and LJM designed the study, MI designed and performed the model simulations, RGB assisted with model setup.

MI wrote the manuscript with feedback, guidance and edits from LJM, PR, SDH, RGB. All authors provided feedback and guidance on the project as it progressed. LJM supervised the project.

**8. Competing Interests**

The authors declare that they have no conflict of interest.

**9. Acknowledgements**

This work was funded by the US National Oceanic and Atmospheric Association (NOAA) via the NOS/NCCOS/CRP Ecological Effects of Sea-Level Rise Program (grant no. NA15NOS4780172) to P.R., S.D.H., and L.J.M as well as the Preston Jones and Mary Elizabeth Frances Dean Martin Fellowship Fund, and by the Virginia Coast Reserve Long-Term Ecological Research Program (National Science Foundation DEB-1832221) via a subaward to the University of North Carolina at Chapel Hill. The authors thank the two anonymous reviewers for helpful comments that improved

the manuscript.

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

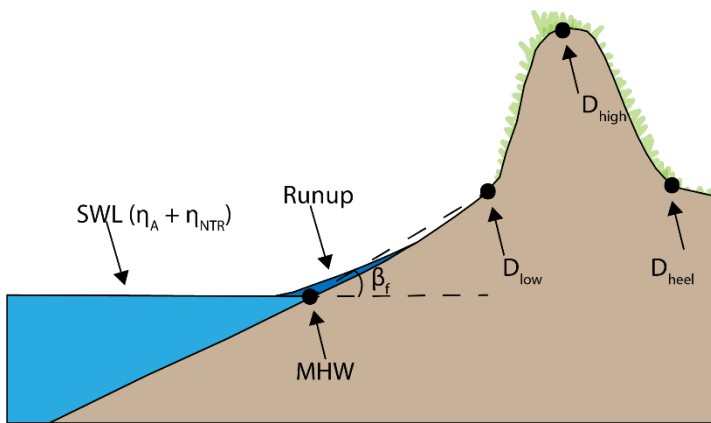

**Figure 1: Schematic beach and dune profile showing the dune toe ($D_{low}$), crest ($D_{high}$), and heel ($D_{heel}$), beach slope ($\beta f$), mean high water contour (MHW), still water level (SWL) and runup, modified from Sallenger (2000).**

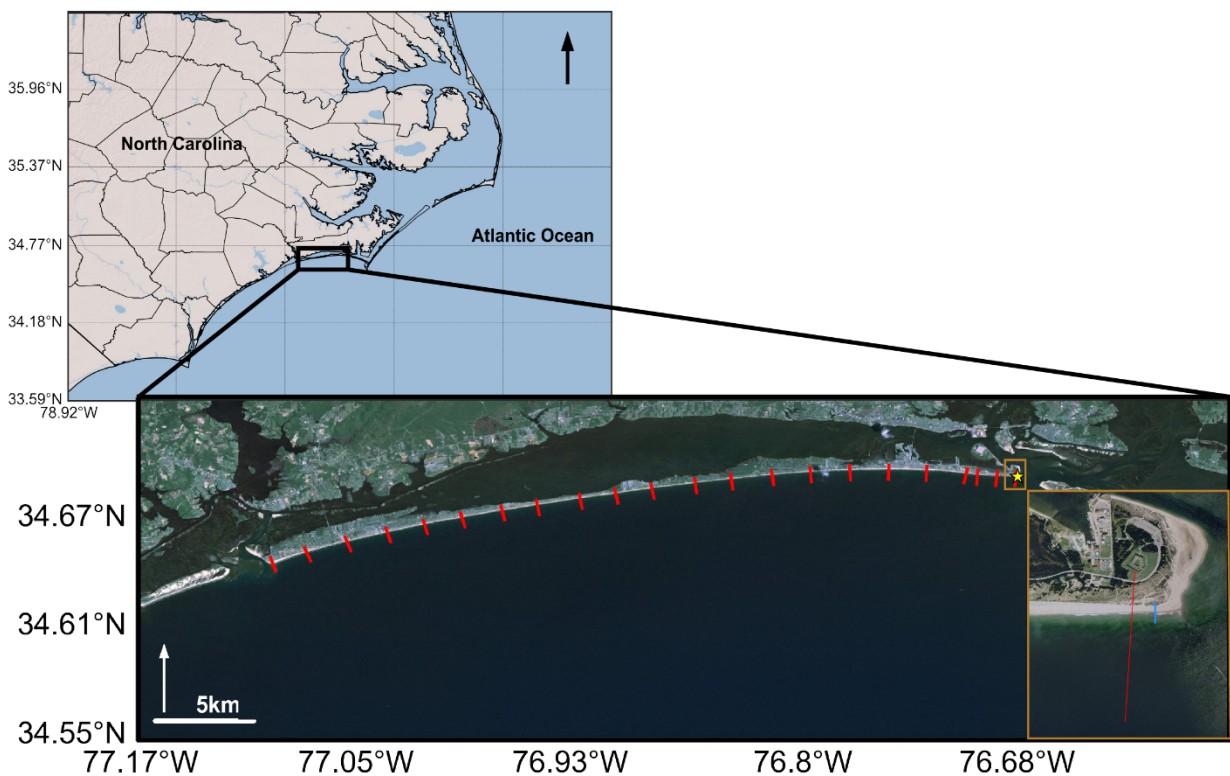

**Figure 2: Map of Bogue Banks, North Carolina showing the locations of the field transects used in this study (red) and the location of the LiDAR profile (yellow star) used to construct the synthetic dune profiles used in this study. The inset image shows the eastern half of Fort Macon State Park with the LiDAR profile location in blue. Figure modified from Itzkin et al. (2020).**

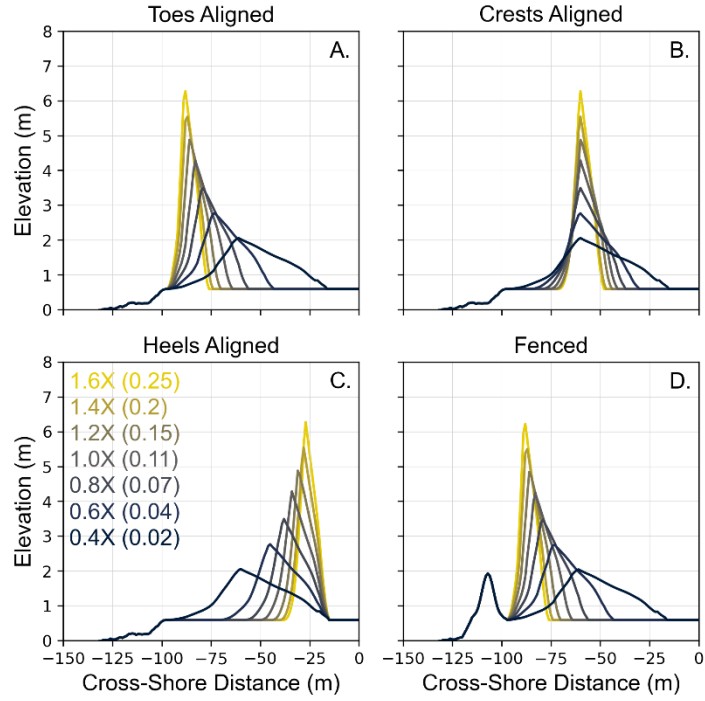


**Figure 3: Synthetic dune profiles; (A) dune toes aligned, (B) dune crests aligned, (C) dune heels aligned, and (D) fenced dune included. The proportional change in aspect ratio (i.e., dune height divided by width) relative to the reference profile (1.0x) is shown in panel C (aspect ratio in parentheses).**

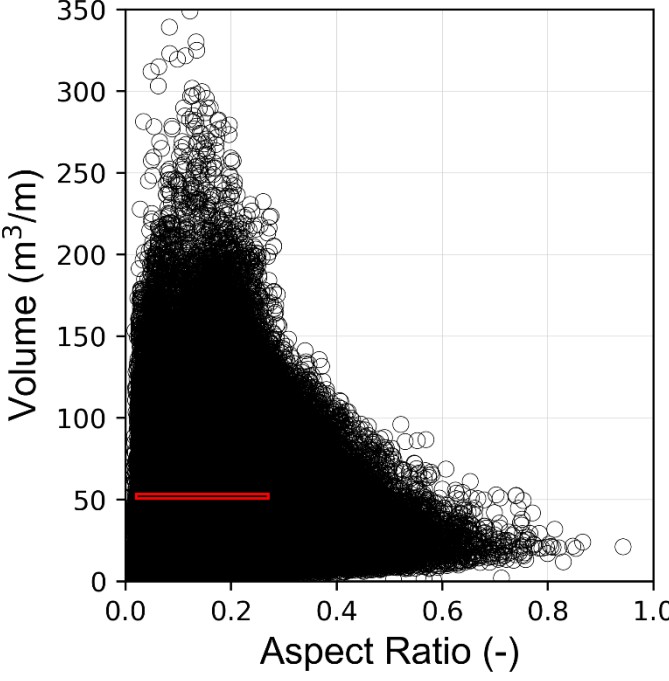

**Figure 4: Dune volume versus dune aspect ratio measured from LiDAR profiles from Bogue Banks, NC, USA, collected between 1997-2016. The red box represents the range of conditions simulated in this study.**

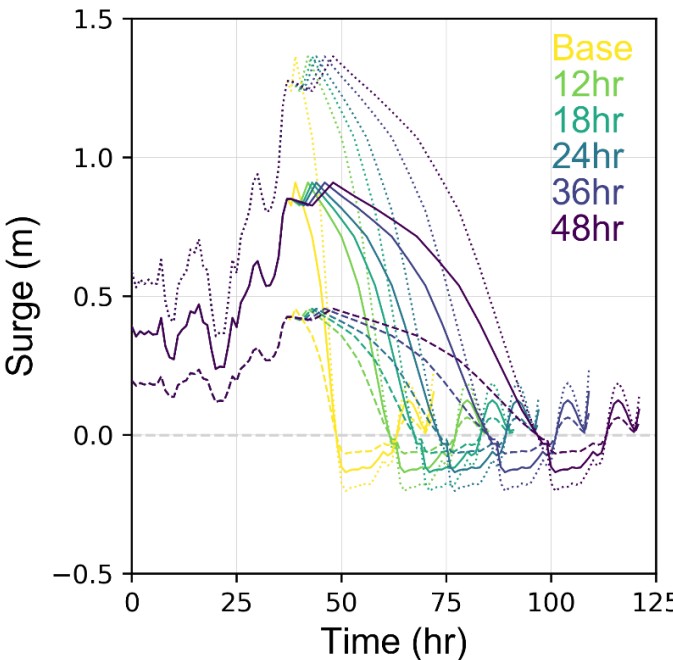

**Figure 5: Synthetic storm surge time series used in this study. Colors refer to the storm duration increase. Dashed lines represent the 0.5x surge, solid lines represent the 1.0x surge, and dotted lines represent the 1.5x surge.**

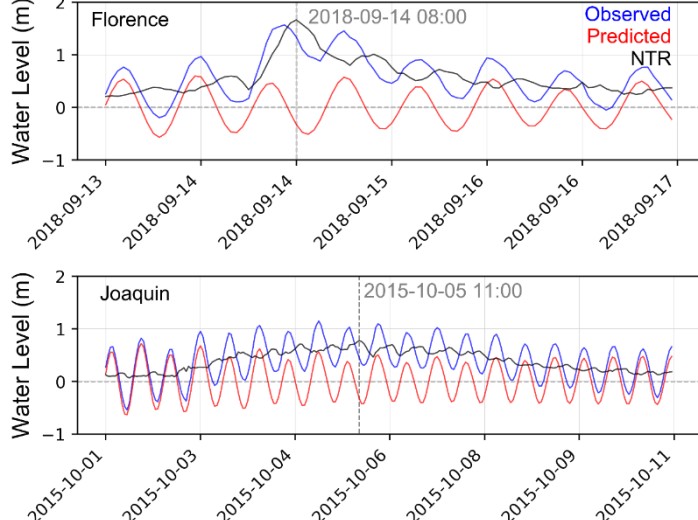

**Figure 6: Hydrographs for Hurricane Florence (top) and Tropical Storm Joaquin (bottom) showing observed water levels (blue), predicted water levels (red), and the non-tidal residual (NTR, black). The timing of peak surge for each storm is highlighted by the vertical dashed line.**

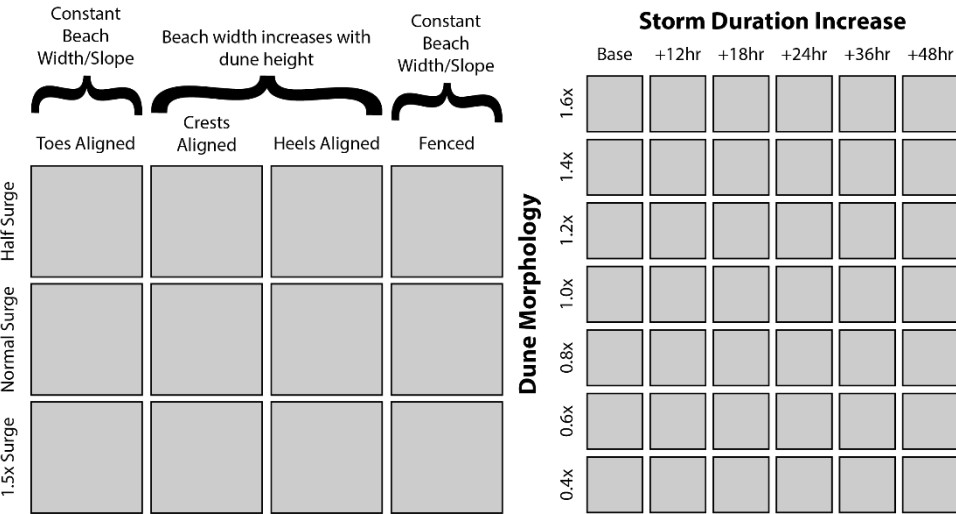

**Figure 7: Schematic overview of XBeach simulations. For every combination of storm intensity and dune alignment shown in the left matrix (12 total), we ran every combination of dune shape and storm duration shown in the right matrix (42 total).**

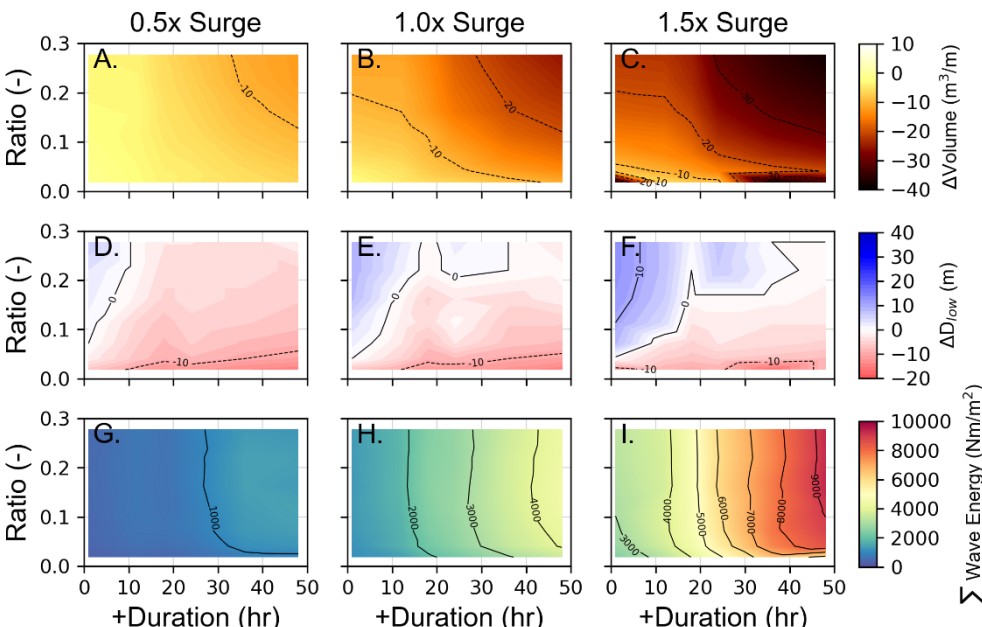

**Figure 8: Dune erosion and wave impact energy for all toe-aligned simulations as a function of aspect ratio versus storm duration. Each column represents a different storm surge level (increasing left to right). The top row (A, B, C) shows the change in dune volume, the middle row (D, E, F) shows the change in dune toe position (negative values indicate landward erosion), and the bottom row (G, H, I) shows the cumulative wave energy impacting the dune.**

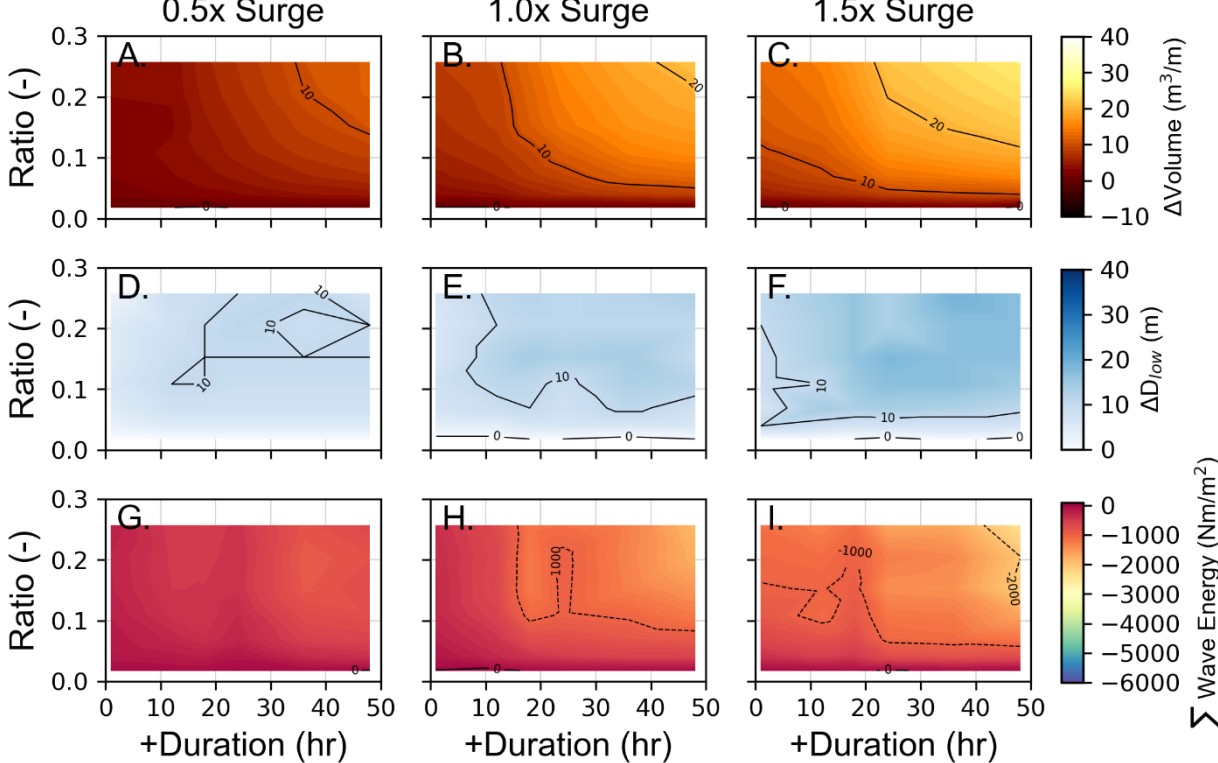

**Figure 9: Relative dune erosion and wave impact energy for all crest-aligned simulations as a function of aspect ratio versus storm duration. The values from the equivalent simulations with the dune toes have been subtracted from the crest-aligned simulations to remove the influence from the varying beach widths in the crest-aligned simulations. Each column represents a different storm surge level (increasing left to right). These values represent a comparison relative to the toes aligned simulation (where beach width is controlled for) such that the top row (A, B, C) shows the amount of volume loss prevented by the wider beach in these simulations, the middle row (D, E, F) shows the additional dune toe progradation induced by the wider beach width, and the bottom row (G, H, I) shows the reduction in wave energy reaching the dune due to the wider (and thus lower sloping) beach.**

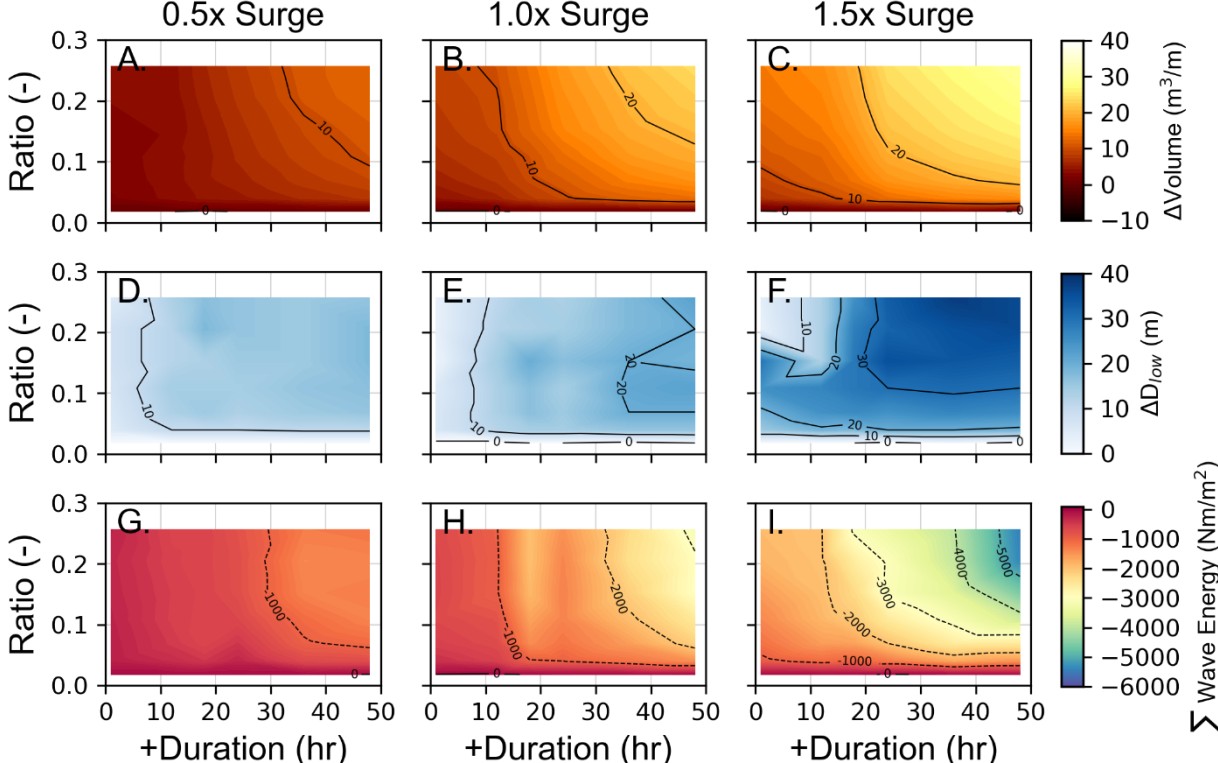

**Figure 10: Relative dune erosion and wave impact energy for all heel-aligned simulations as a function of aspect ratio versus storm duration. The values from the equivalent simulations with the dune toes have been subtracted from the heel-aligned simulations to remove the influence from the varying beach widths in the heel-aligned simulations. Each column represents a different storm surge level (increasing left to right). These values represent a comparison relative to the toe-aligned simulation (where beach width is controlled for) such that the top row (A, B, C) shows the amount of volume loss prevented by the wider beach in these simulations, the middle row (D, E, F) shows the increase in dune toe progradation induced by the wider beach width, and the bottom row (G, H, I) shows the reduction in wave energy reaching the dune due to the wider (and thus lower sloping) beach.**

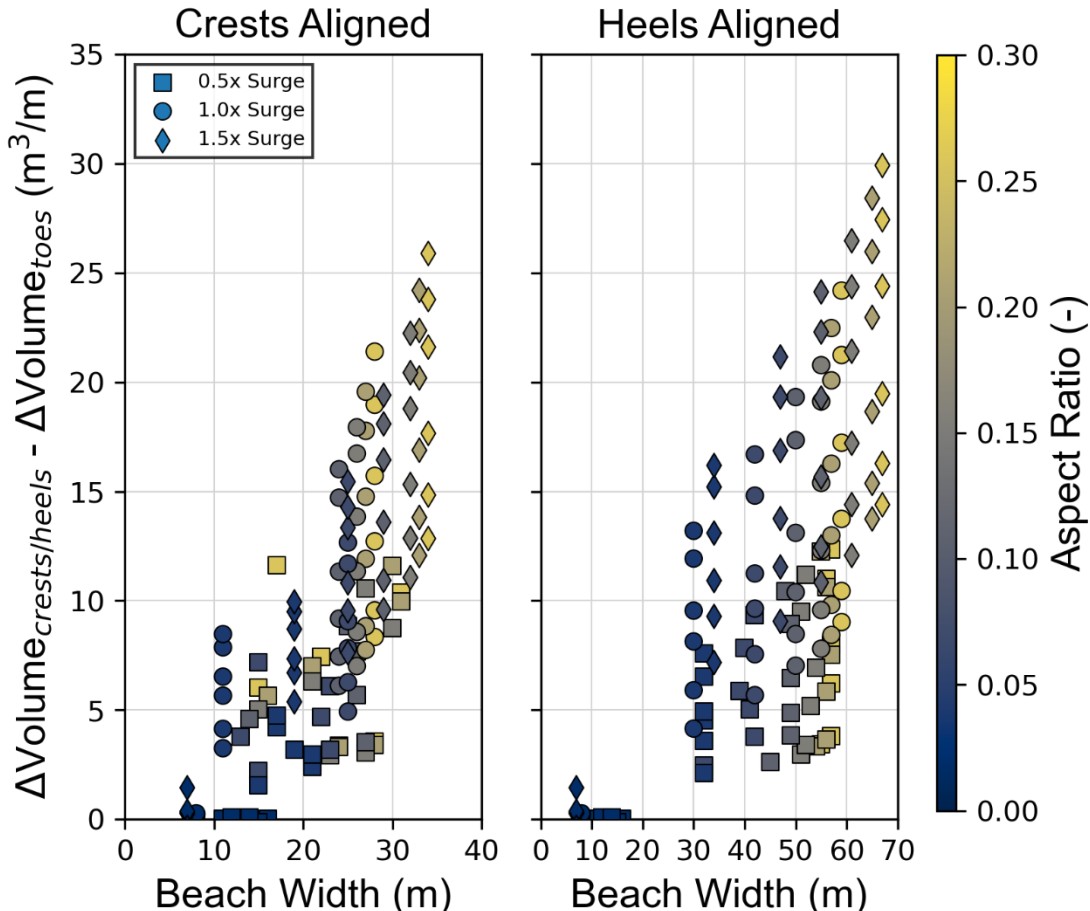

**Figure 11. Volume loss from the crest- and heel-aligned simulations minus volume loss from the equivalent toe-**
**aligned scenarios versus the initial beach width for the crest- and heel-aligned simulations. The color**
**corresponds to the dune aspect ratio and the shape corresponds to the storm surge level.**

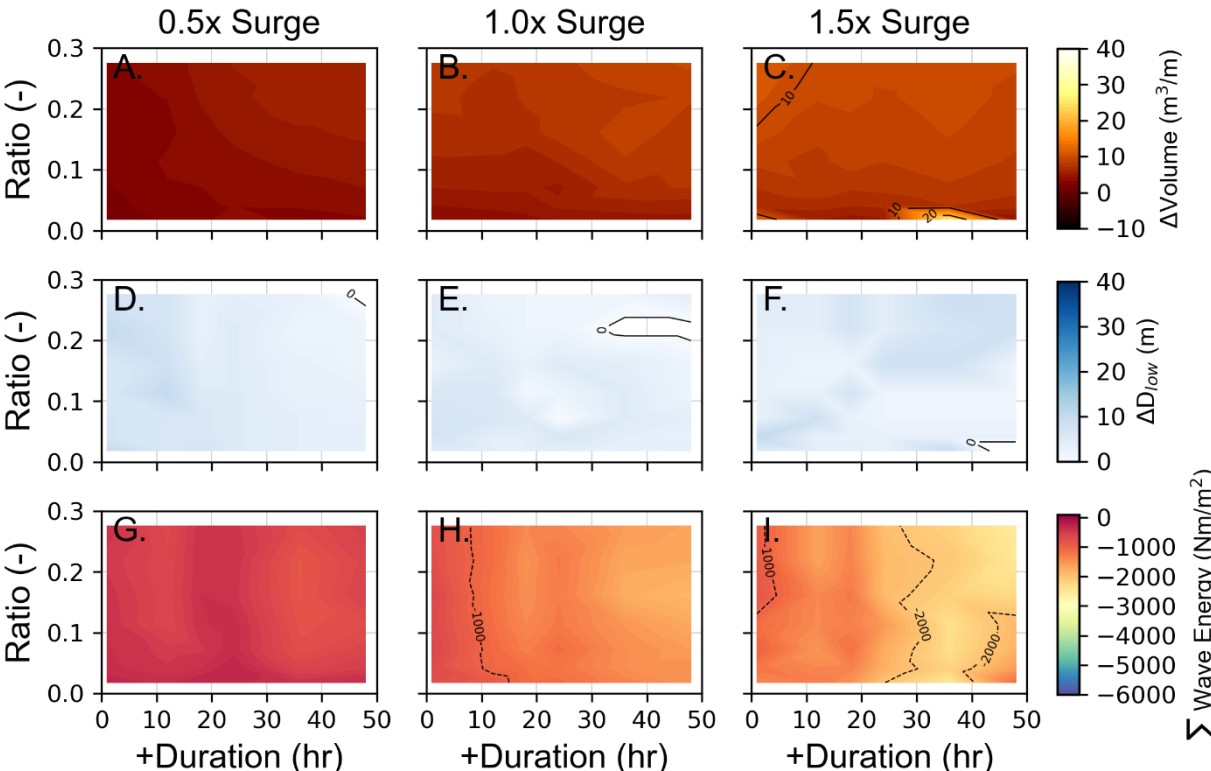

**Figure 12: Relative dune erosion and wave impact energy for all fenced simulations as a function of aspect ratio versus storm duration. The values from the equivalent simulations with the dune toes have been subtracted from the fenced simulations in order to remove the influence from the presence of the fenced dune seaward of the natural dune. Each column represents a different storm surge level (increasing left to right). These values represent a comparison relative to the toe-aligned simulation (where beach width is controlled for and there isn't a fenced dune) such that the top row (A, B, C) shows the amount of volume loss prevented by the fenced dune in these simulations, the middle row (D, E, F) shows the increase in dune toe progradation induced by the fenced dune, and the bottom row (G, H, I) shows the reduction in wave energy reaching the dune due to the fenced dune.**

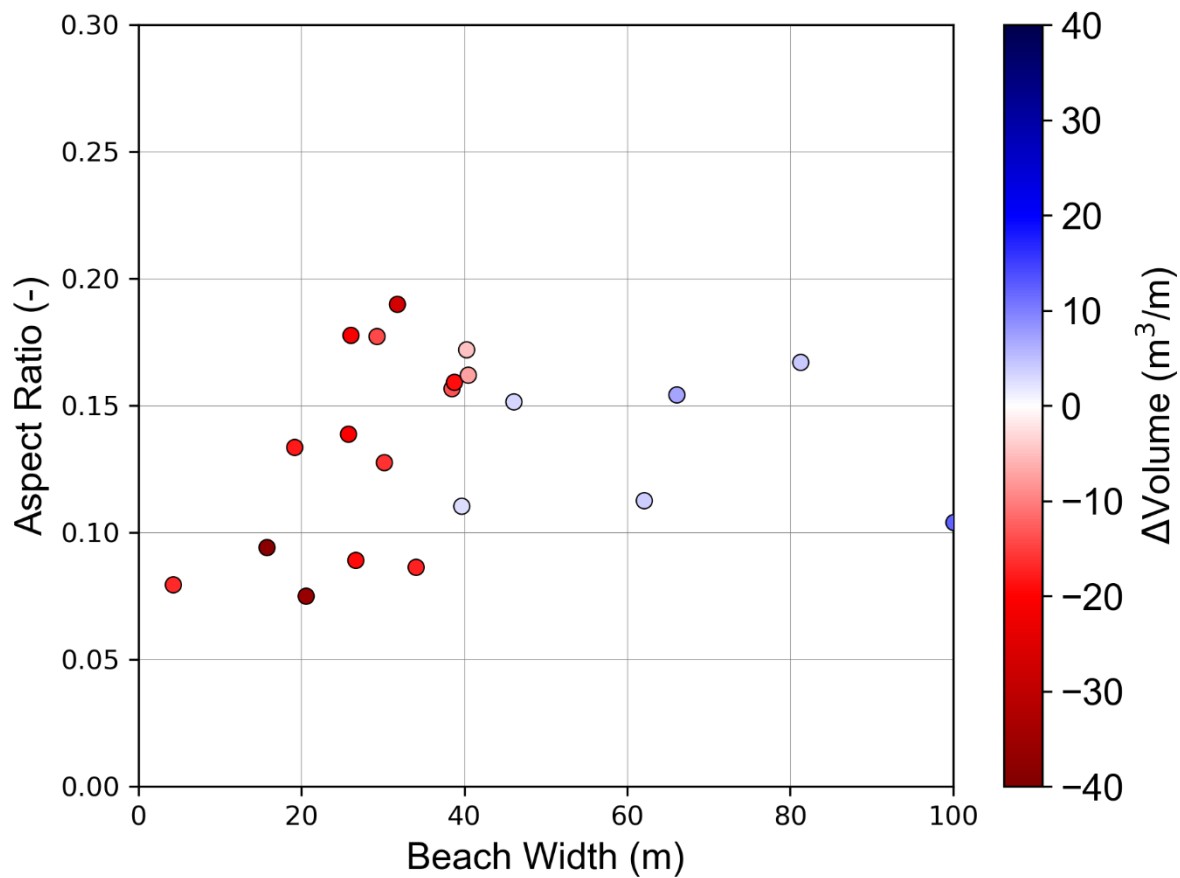

**Figure 13. Pre-storm aspect ratio versus pre-storm beach width for foredunes at Bogue Banks, NC. Each point represents a transect and points are colored based on the change in volume from 2017-2018.**

|  | Aspect Ratio (-) | Volume (m3/m) | Dune Height (m) | Dune Width (m) | Dune Slope (m/m) |
|---|---|---|---|---|---|
| **1.6X** | 0.25 | 52.75 | 5.68 | 49 | 0.15 |
| **1.4X** | 0.2 | 50.64 | 4.95 | 50 | 0.13 |
| **1.2X** | 0.15 | 51.76 | 4.29 | 53 | 0.12 |
| **1.0X** | 0.11 | 53.27 | 3.69 | 56 | 0.1 |
| **0.8X** | 0.07 | 53.45 | 2.9 | 60 | 0.08 |
| **0.6X** | 0.04 | 53.25 | 2.18 | 67 | 0.06 |
| **0.4X** | 0.02 | 53.29 | 1.47 | 82 | 0.04 |

**Table 1. Initial dune shape parameters for synthetic dune profiles used in this study. Note that dune height is measured from the dune crest to the dune toe, the initial dune toe elevation is constant at 0.59m for all profiles. The values in the left column refer to the modification to the reference dune profile as described in equations 1 and 2 and the profiles in Figure 3.**

660

| Parameter | Default Value | Used Value |
|---|---|---|
| break | roelvink2 | roelvink_daly |
| gamma | 0.55 | 0.52 |
| eps | 0.005 | 0.1 |
| facSk | 0.1 | 0.15 |
| dryslp | 1 | 4 |
| hmin | 0.2 | 0.01 |

**Table 2. Non-default XBeach parameterizations used in erosion simulations.**