# Peer review of "The Relative Influence of Dune Aspect Ratio and Beach Width on Dune Erosion as a function of Storm Duration and Surge Level"

_Earth Surface Dynamics, 2020_

## Referee Comment (RC1) · Anonymous Referee #1 · 16 Dec 2020

The manuscript under discussion here has as major purpose to analyse dune erosion as a function of dune aspect ratio (i.e., dune height versus dune width) for storms of varying intensity and duration by simulating hydrodynamic processes, sediment transport, and morphologic change. For that, the authors create a series of synthetic dunes to run a series of sensitivity analysis.

The manuscript is well organized, easy to read and clearly presents its objectives. However, I am afraid that the outcomes from the author's experiments do not actually support their conclusions, and more important, are not the best to address the original objective. The conclusions reached by the authors overlap very well grounded facts,

namely the importance of the beach width as a major control of dune erosion, stated by several earlier works; e.g. review works (Davidson, Hesp and Miot da Silva, 2020) and works based on field observations (e.g. Burroughst and Tebbens, 2008; Charbonneau et al., 2017; Claudino-sales et al., 2008; Crapoulet et al., 2017; Galiforni Silva et al., 2019; Héquette et al., 2019; Itzkin et al., 2020; Keijsers et al., 2014; Pye and Blott, 2016), making it difficult to understand what is the actual contribution of the manuscript other than calling our attention to the fact that dunes may erode over time (usually not by a single storm), reducing their capacity to prevent overwash and inundation, depending not only on their elevation but also on their width. I believe that it would make more sense to me to call our attention to the fact that the shape of the dune, not only the aspect ratio as that might be a bit limiting indicator as the shape of a dune can vary very much, turning it very important to consider indicators that inform about the volume, in addition of course, to the elevation, as that is the key parameter that determines the impact regime, the impact can shift to overwash or inundation if a particular height is maintained over a certain width of the dune, but this is not informed by the aspect ratio. So, I would say that additional information other than only the aspect ratio would be needed to actually understand if a particular dune can cope with the impact of one or several storms if the shoreline is retreating (or if the dune is being eroded) the aspect ratio of the dune may also change, as they are all but very irregular features.

In this line, the authors state that the amount of dune erosion or the vulnerability of coastal dunes does not only depend on dune height but also on its width, as dunes can also erode by collision regimes, and thus, an aspect ratio that includes both dimensions should be used instead of only the elevation. In addition, they also state that dune erosion might be influenced by this aspect ratio. In general, I do agree with the hypothesis stated by the authors, however, I cannot fully agree with the approach used to support their statements and reinforce my concern regarding the originality of the contribution from this work. My main concern is linked to the experiments chosen by the authors, namely to the synthetic dunes used and the way the authors have decided

to create the four different configurations, which ended up having dunes with different morphologies (symmetric, non-symmetrical and with changing front dune slopes) and different distances to the shoreline that cannot be easily compared. In fact, the results suggest other factors might be more important than the dune aspect; namely the beach width, which determines the level of impact of the storm over the dune.

I will try to synthesize my concerns focusing on some statements from the abstract of the manuscript and mainly linked to the results of the experiments or simulations.

The authors state in the abstract that "low aspect ratio (low and wide) dunes lose less volume than high aspect ratio (tall and narrow) dunes during longer storms, especially if they are fronted by a narrow beach". Regarding the first part of the sentence and looking at Figure 7, where the results from the simulations are presented, it is not so obvious this affirmation as low aspect ratio dunes only erode less when using the fixed dune toe configuration (narrowing the size of the beach). The higher aspect ratios in this case are related to dunes with very steep seaward slopes resembling scarped dunes. As the authors state, and following Hesp 1988, these dunes can be more unstable and have greater probabilities of crest collapse, is all the sediment from the collapsed dune removed by the waves? Having in mind the main principles of dune/beach erosion, the amount of volume eroded should not be very different or dependent on the dune shape, but on the volume of sand that needs to be eroded as it is a response to adjust the beach profile to more energetic waves. It is not so obvious from these results that low ratio dunes erode less, as the authors change the shape of the dunes and their distance to the shore, and therefore, they cannot be easily compared anymore. In this regard, I would imagine that a more synthetic dune would help and allow direct comparisons. By a more synthetic dune I imagine something that preserves the slopes (shape) changing the ratio (even though not so realistic, a cube would make it easier). In fact, when looking again at figure 7, the lower ratio dunes are eroding more for all cases but the toe fixed one, which is due to a change in the seaward slope of the dune and not merely to a change in the ratio, as those have very very gentle slopes that

do not make easy to identify which is the actual location of the dune toe (or transition to the beach). So, from what is shown here, high ratio dunes are not "specially" losing more volume of sand when the beach is narrow as the authors state, but "only" when the beach is narrow, otherwise they almost don't lose sand. The latter results from the fact that the other configurations (fixed crest and heel) present wider beaches and so, they may enter the collision regime later, or not at all. The authors also suggest that this affirmation mostly applies to longer storms. However, from figure 7f, j, g, k, we see that the erosion increases over time also for low aspect dunes if the configuration of the dune is different, so, again, the problem of comparing dunes with different shapes and distance to the shore does not help interpreting the outcomes of this work, and to assess the role of the ratio.

Also in the abstract, the authors affirm: "During more intense storms, low aspect ratio dunes experience greater erosion as they are more easily overtopped". I see again a problem when comparing dunes with different shapes and distances to the shoreline. The affirmation sounds totally logic to me as the amount of volume needed to erode depends on the magnitude of the storm, and as stated by previous authors: "one of the most significant factors affecting the magnitude of spatial and temporal change to a foredune during an erosion event is the height of the mean water level during the storm (Davidson, Hesp and Miot da Silva, 2020 and references therein)", and in fact this is the main factor that the authors assess when changing the magnitude of the event, the water level. However, when looking again at figure 7 (namely 7i), high and low aspect ratios share maximum volume loses, and in fact the main difference with fig.7e, is the fact that a higher water level reaches the low aspect ratio dune crest more easily than low water levels. Conversely, low aspect ratio dunes for the configurations other than the fixed toe one, are more vulnerable because their toe is closer to the shoreline when compared to the high aspect dunes, which are far from the shore in the crest and heel fixed configurations, which in turn makes it difficult to compare volumes of erosion.

Still in the abstract, the authors state that in managed scenarios (by managed dunes,

they refer to those sites where a fenced dune is constructed seaward of the existing natural dune) a fenced dune effectively prevents the natural dune behind from experiencing volume loss. Again, this is what should be expected as having a fenced dune implies in their case that the shoreline is again seaward and distant from the toe of the natural dune, resembling a wide backshore. Yet, the authors mention that the volume loss can be reduced up to 50%, which is a number that depends on their experiments, as if they have built a larger fenced dune it would be greater, also if instead of using the toe fixed they had used the crest or heel fixed configurations, that % would increase. So, again, it is not clear the point/contribution of the authors as these are well grounded ideas/facts usually used for the managers to design the actions to take.

In this line, the authors also state: "a wide beach offers the greatest protection from erosion in all circumstances regardless of dune morphology or storm characteristics". This is again, the expected result, so, are the authors trying to convince the readers that XBeach can simulate erosion? And yet, this is again something that many examples in the literature have demonstrated. Finally, and yet in the abstract, the authors end saying: "in maintaining wide beaches and dunes, the protection offered in the short-term must be considered against long-term detrimental effects of potentially limiting overwash fluxes, which are critical to maintaining island elevation as sea level rises". This idea is developed at the end of the discussion section. From what I understand, a nourished beach, if using the adequate sediments, could provide the needed sediments for the dune to cope with sea-level rise as that is the regular mode dunes grow vertically. The authors claim that maintaining large beaches would prevent rollover and thus, the natural adaptation of the barrier to sea-level rise. Here, I cannot agree with the authors because barriers, if sediment input is enough (so they maintain their beach, even if artificially) can adapt by adapting the vertical elevation of the dune feed from the abundant sediments of the beach, and not only through rollover, right? I see more the problem of being able of maintaining large beaches with the adequate sand and in terms of costs. In this context, if sea level rises, so will the nourished beach berm and thus the dune, unless winds are not efficient anymore, which seems not to be the

case in the future. Therefore, I would recommend the authors to better explain what the actual message is that they try to highlight here and to state what is their opinion about the possible coupled adaptation of the dune in that context.

More specific comments: Regarding the introduction, I would expect here that the authors present the main factors controlling the magnitude of dune erosion by storms, as this will be the main point of discussion, also considering the scales of change, from a single storm to a series of storms or the time-scale of relevance for managing would be interesting. In relation to the methods, they mention that the effect of the configuration of the beach, namely the beach slope, will be also assessed, however, this is not thereafter clearly evaluated. In line 281, I would add here the importance of the width of the dune to the impact of storm groups also.

---

## Referee Comment (RC2) · Anonymous Referee #2 · 18 Dec 2020

The Influence of Dune Aspect Ratio, Beach Width and Storm Characteristics on Dune Erosion for Managed and Unmanaged Beaches By: Michael Itzkin et al.

Submitted to Earth Surface Dynamics

Overview: This paper presents new XBeach modelling to examine the impacts of dune aspect ratio and beach width on the modelled erosion volumes for a set of different storms (including varying duration and surge levels). The three main objectives of the paper were: (1) How does storm duration affect volumetric dune erosion as a function of foredune aspect ratio? (2) How do variations in storm TWL affect volumetric dune erosion as a function of foredune aspect ratio? (3) How does the morphology of the

beach (i.e., width and slope) affect volumetric dune erosion independent of foredune aspect ratio.

Overall, I found the paper well written and quite succinct. From a point of view on was there a substantial increase in our fundamental understanding into dune erosion, I was less convinced. Not much in the paper surprised me or told me something I didn't know, but more reaffirmed my understanding/observations/past work. That's not to say that more couldn't be presented to improve the paper and provide further understanding that I think is unique to modelling work as you have high resolution results that you can interrogate more than you have presented here. By presenting more and digging more into the results I think you could better answer your three objectives above.

For example: Your dune profiles were very different and in XBeach, erosion occurs if a cell is determined to have been 'wet' so since your higher aspect ratio dunes had more sand closer to the dune toe, they would expect to have more erosion volumes by the nature of the model and not necessarily by a physical meaning. XBeach dune erosion is purely ad-hoc. If a cell is wet, it compares it to your wet slope and erodes it if it's above this critical value. Realignment can also take place if dryslp is exceeded. None of this is really based on physics of dune erosion. Dunes hold much larger scarps under active erosion (See Palmsten and Holman paper for examples but many others as well including work by Erikson and Hanson -> dune notching paper , Larson Erikson and Hanson (2004) and all the work on dune impact models (Overton et al) all show this). The sand is typically (from my experience using XBeach) also immediately moved offshore (to keep the wetslp low) so the feedback mechanisms we'd see in real erosion are not there where slumped sand protects the dune toe. The model has limitations and I can accept those but I think you need to acknowledge them a bit more here and realize what we can (and cannot) learn from these results.

Consider the very different dune aspect ratios you are considering and the distribution of sand in the cross-shore, it would be good to see dune toe recession presented as well as you refer to volumes (which I also think are needed) but when you align toe,

heal, center, and with each of the aspect ratios you change the distribution of the volume in the dune. So small events will erode a lot when the toe is aligned because there is a lot of sand up close, but dx (dune toe erosion) might be similar and this is a key variable of interest to engineers/managers. The model is a grid so you are 'eating away' at the dune 1 grid point at a time as a function of the predicted TWL. Default dry slopes in XBeach are also quite flat compared to what would be capable in active dune erosion (see for example lab studies of Palmsten and Holman 2012, https://www.sciencedirect.com/science/article/pii/S0378383911001633; Palmsten and Holman 2011, https://agupubs.onlinelibrary.wiley.com/doi/full/10.1029/2011JC007083; Palmsten and Splinter 2016, https://www.sciencedirect.com/science/article/pii/S037838391600017X - this latter one explicitly looked at XBeach and my memory is that to match the lab data they used dryslp almost 4x the default value to allow for near vertical scarping)

Can you also answer your objectives in terms of dune toe recession (as well as volume) to get a deeper understanding/picture of how dune aspect ratio effects overall erosion. One would expect that perhaps that higher aspect ratios might also have less dune toe recession as more sand is dumped onto the beach and may offer protection.

I would also like to see plots of beach width change over the storm. This is something you say is quite important to your results – wider beaches offer more protection. Something that other researchers have also shown to be quite important (eg. Plant and Stockdon, 2012. Probabilistic prediction of barrier-island response to hurricanes https://agupubs.onlinelibrary.wiley.com/doi/full/10.1029/2011JF002326; Beuzen et al. 2019. Controls of Variability in Berm and Dune Storm Erosion https://agupubs.onlinelibrary.wiley.com/doi/abs/10.1029/2019JF005184) Beach width (or safe corridor width) is also a key parameter that engineers/managers are wanting. How does your beach width over a storm impact on the erosion – does it need to be completely removed or only a certain percent for the dunes are vulnerable.

I think if you could present your results looking at multiple parameters (volume, dune to retreat, beach width change, dune impact hours) then the reader would get a

much richer understanding of the impacts these changes to dune aspect ratio/beach width/storm duration had on the study. Volumes themselves only tell a small part of the story.

Other Scientific Aspects to be considered: L35: "Considering that wave runup is most likely to impact the dune face (i.e., collision; Sallenger, 2000)—which is more likely to affect the width of the dune rather than the height—is the most temporally common impact regime during a storm (Brodie et al., 2019; Stockdon et al., 2007), the width of the dune is an important predictor of how much erosion a dune might experience during a storm." I find this sentence really hard to read. Consider revising. As well, width won't be a predictor so much of the amount of erosion I would think, but of the erosive vulnerability of the dune itself. This paper might be of interest to you as it looks at both dune characteristics (height/width) and beach width in terms of erosion and flooding risks in storms: Leaman et al. (preprint, under review in Coastal Eng). A Storm Hazard Matrix combining coastal flooding and beach erosion. https://eartharxiv.org/repository/view/1753/

L171: I am a bit concerned about leaving all other XBeach parameters as default as many studies have shown this isn't appropriate outside of the highly dissipative beaches for which the model was originally designed (along the Dutch coast). Leaving all other parameters as default has implications between overwash and collision regime erosion estimates as noted by previous researchers such as Passeri et al. and Simmons et al.. Not accounting for these processes will impact on your results. Why weren't these considered?, even is the cases were limited where overwash did occur? Others have also shown sensitivity of the erosion to parameters. Eg references below (note this isn't a complete list, just ones I could think of off-hand). Passeri et al. The influence of bed friction variability due to land cover on storm-driven barrier island morphodynamics https://www.sciencedirect.com/science/article/pii/S0378383917301114 Simmons et al. Calibrating and assessing uncertainty in coastal numerical models https://www.sciencedirect.com/science/article/pii/S0378383916303234#f0030

[Figure]

Splinter and Palmsten. Modeling dune response to an East Coast Low
https://www.sciencedirect.com/science/article/pii/S0025322712002034

L183: "or when dunes are located closer to the shoreline (represented by the dune toes-aligned scenarios; Figure 7)." I am a bit confused by this as the effect of beach width would be shown not when the dune toes were aligned (and all beaches had the same beach width) but instead when the dunes were aligned at their crest or heel, which then changes their beach width. Ideally you should be comparing the cases for the same dune aspect ratio at these three positions to determine if effect of beach width. And this is repeated for each of the dune aspect ratios. This would be an interesting thing to see in my opinion (same dune aspect ratio plotted for the 3 positions within your dune toe, heal, crest align) to see how BW effects erosion for the same dune.

Wider beaches offer a big buffer of sand that must be eroded before the wave action can get to the dune and frictional damping of the runup would also occur, lessening the probability of a dune experiencing wave impacts. Looking at dune impact hours could be interesting and provide some good insight here.

L185: "situated farther from the shoreline (dune heels-aligned)" as above, I don't see how having the dune heels aligned also indicates they are further from the shorelines as each of these cases would have a different beach width.

L241: "Additionally, the sensitivity of the dune to decreases in storm duration was inversely proportional to the beach width such that dunes fronted by wide beaches were noticeably less sensitive to increases in storm duration than dunes fronted by narrow beaches (Figure 9)." – It would be great to see figures that show beach width change over the storm.

Specific Minor Editorial Comments: L75: 'aspect ratio' is repeated twice

L91: replace 'Dtoe' with 'Dlow' to match figure 1 and to remove confusion as I believe

that Dlow=Dtoe.

L95: "Given that Dlow was held constant across all simulations" I think should be "Given that prestorm Dlow was held constant across all simulations".

L147: remove '.' in 'approximately.'

Overall, I think the paper could be improved to provide a fuller understanding of the complexities of dune erosion and how dune aspect ratio, beach width and storm duration/intensity impact on the model results. I have provided a number of example references to consider, but I'd like to acknowledge here that these are limited to what I could recall off hand rather than providing a complete list of relevant resources. Please consider these as examples and you might find more suitable ones within these papers as well.

---

## Editor Comment (EC1) · Andreas Baas (Editor) · 23 Feb 2021

Dear Authors,

Thank you for your patience awaiting the reviews of your manuscript. The third referee suffered an arm injury and was unable to write up a full review report but did send me via direct email a number of general comments and suggestions.

The first two review reports both identify some major limitations of the work presented here. Of key concern are the lack of novelty of the findings and inconsistencies between the objectives, simulation scenarios, and conclusions. These concerns are

shared by the third referee, whose comments I can paraphrase here as: /- Conclusions are stretched, and the impact of beach width on dune erosion is already well-known. It is also obvious that dune erosion is less when an artificial dune is placed in front; the % reductions found are purely a consequence of the arbitrary geometry of the artificial dune. /- Systematic exploration of dune erosion is a good idea, but the set of scenarios is not optimal and there is little interrogation of the details of the erosion process itself. /- There is thus more to be found in analysis of the simulations, perhaps also regarding the parametrisations in Xbeach. Figure 7 shows some interesting features that are not fully discussed. /- The 'managed dune' set-up is rather limited, it should include modifications of the dune toe such as marram planting, sand ramps, and sand fencing, interventions that aim to widen an existing/eroded dune again.

Based on the views of the three referees I believe the work requires a major revision before it can be considered again. Such a revision will likely involve further/new simulation work and re-analysis (referee #2 in particular provides a number of suggestions). A new manuscript presenting this work will be sent out for review again.

Andreas Baas, Handling Associate Editor

––––––––––––––––––––––––––––

---

## Author Comment (AC1) · 14 Apr 2021

We appreciate this reviewer's thoughtful comments, which have assisted us in improving the manuscript. In what follows, we respond (in non-bold text) to reviewer's comments (shown in bold text). New or revised manuscript text is italicized. We found it simplest to revise the manuscript at the same time that we prepared our response to reviewer comments. Thus, the line numbers refer to the submitted version unless otherwise noted and indicate where new text has been inserted. Note that all revised Figures referred to throughout the response can be found at the end of this document.

Here, we provide a brief summary of the major changes we have made in response to Reviewer 1: We now more carefully control for the effects of dune aspect ratio and beach width in our analyses. We also clarify our goals and approach with respect to these points throughout the manuscript. To accomplish this, we have reframed our presentation, restructuring the results section and reorganizing the discussion section to highlight that 1) We isolate the effects of dune aspect ratio on dune erosion through analysis of our toes-aligned simulations (in which beach width is held constant). These scenarios allow us to also examine the effects of storm duration and storm surge level on dune erosion as a function of dune aspect ratio. 2) We then isolate the effects of beach width on dune erosion by subtracting the effect of dune aspect ratio on dune erosion as determined from the corresponding toes-aligned simulations, and 3) Our fenced-aligned simulations then provide additional insights into how dune erosion is influenced by emplacement of sand fences as a function of storm duration and surge level. We also propose to change our title from "The influence of dune aspect ratio, beach width and storm characteristics on dune erosion for managed and unmanaged beaches" to- "The relative influence of dune aspect ratio and beach width on dune erosion as a function of storm duration and surge level." Consistent with this revised and more clarifying approach, we adjusted the figures and their presentation in the following way:

- Figure 7 now only shows results from the toes-aligned simulations (instead of all four configurations). There are now three columns (one for each surge scenario), and the top row shows the volume loss, the middle row shows dune toe migration, and the bottom row shows the wave energy reaching the dune (these are new metrics provided in response to comments from Reviewer 2). We include equivalent figures (Figures 8, 9, 11) for the crests-aligned, heels-aligned, and fenced simulations but with the influence of beach width removed by subtracting the toes-aligned results. We have removed original figures 8 and 9.
- A new figure (Figure 10) now shows the volume loss for the crests- and heels-aligned simulations minus the volume loss from the equivalent toes-aligned simulations. These data were plotted versus the initial beach widths of the crests- and heels-aligned simulations to demonstrate the volume loss that was prevented as a function of the varying beach width for these simulations.

Below we respond to specific points in the review although we note the key changes to the manuscript to address the review have been summarized above.

**The manuscript under discussion here has as major purpose to analyse dune erosion as a function of dune aspect ratio (i.e., dune height versus dune width) for storms of varying intensity and duration by simulating hydrodynamic processes, sediment trans-**

**port, and morphologic change. For that, the authors create a series of synthetic dunes to run a series of sensitivity analysis.**

**The manuscript is well organized, easy to read and clearly presents its objectives. However, I am afraid that the outcomes from the author's experiments do not actually support their conclusions, and more important, are not the best to address the original objective. The conclusions reached by the authors overlap very well grounded facts, namely the importance of the beach width as a major control of dune erosion, stated by several earlier works; e.g. review works (Davidson, Hesp and Miot da Silva, 2020) and works based on field observations (e.g. Burroughst and Tebbens, 2008; Charbonneauet al., 2017; Claudino-sales et al., 2008; Crapoulet et al., 2017; Galiforni Silva et al.,2019; Héquette et al., 2019; Itzkin et al., 2020; Keijsers et al., 2014; Pye and Blott,2016), making it difficult to understand what is the actual contribution of the manuscript tother than calling our attention to the fact that dunes may erode over time (usually not by a single storm), reducing their capacity to prevent overwash and inundation, depending not only on their elevation but also on their width.**

We appreciate that the reviewer found the manuscript well organized, easy to ready and to clearly present its objectives—thank you for this positive feedback. We note that the major changes made to the manuscript and summarized on page one of this response improve the connection between objectives stated in the paper, the experiments and the results. In the comment directly above, the reviewer provides several citations to papers that discuss the role of beach width in dune erosion. However, in our reading of these papers while they suggest or state the importance of beach width, they do not quantitatively demonstrate its importance. Not only does our work show that beach width plays a role in dune erosion, it also demonstrates the relative roles of dune morphology (height and width, quantified as the aspect ratio) and beach width in determining how a dune will erode under various storm conditions. Storm impact studies focus on the role of dune morphology (typically just height) but here we quantitatively show, through the use of numerous XBeach simulations, that dune morphology is secondary to beach morphology in predicting storm-induced dune erosion.

Regarding the concern that we are demonstrating something that has already been shown, in the review paper by Davidson et al. (2020)—the most recent of the papers cited by the reviewer— beach width is mentioned as a factor that regulates foredune scarping, and the paper concludes with a statement that calls for just the type of study we have conducted stating that "Further research is required to determine the relative importance of some of the controls on the degree of scarping detailed here, and future investigations on what constitutes the most vulnerable beaches and dune systems, and why, is required" (Davidson et al., 2020) . The paper by Davidson et al. (2020) also states "observations and tests that record the difference in the magnitude or degree of scarping occurring on reflective versus intermediate versus dissipative beaches under the same storm and offshore swell conditions would be very useful". In our revised manuscript wee determine, after incorporating the suggestions from the reviewers, the relative importance of the various controls (TWL, dune height/width, beach width/slope, and storm duration) on dune erosion. Further, while we do not explicitly reference reflective/intermediate/dissipative beaches, we do simulate erosion across a wide spectrum of beach morphologies using "the same storm and offshore swell conditions." Thus, our paper specifically contributes to the further research that Davidson et al. (2020) call for.

The reviewer also suggests that the main message of the paper is that that "dunes may erode over time (usually not by a single storm), reducing their capacity to prevent overwash and inundation, depending not only on their elevation but also on their width." We think this is an oversimplified statement of our study. In response, we have reviewed the text carefully and made changes that we believe helps better communicate our findings to the reader (revised manuscript lines 77-86 and 437-439 respectively):

*"The main goal of the work presented here is to assess how dunes erode during a single storm, as a function of dune aspect ratio (i.e., the combined measure of height and width), beach width, and management efforts that led to the formation of an artificial dune (i.e., sand fences). While previous studies of storm impacts on dunes have primarily focused on dune height (e.g., Long et al., 2014; Sallenger, 2000; Stockdon et al., 2007), recent studies suggest that the dune width may also be a key predictor for how much dune erosion will be experienced* (Leaman et al., 2020). *Additionally, although beach width has been posited as a strong predictor of dune erosion (e.g., Burroughs and Tebbens, 2008; Claudino-Sales et al., 2008; Itzkin et al., 2020; Silva et al., 2018) quantifying and understanding the relative role of beach width in dune erosion processes requires further investigation* (Davidson et al., 2020)*."*

*"In this study we analyzed how coastal foredunes erode during a storm as a function of their aspect ratio (height and width), beach width, and the presence of management interventions (i.e., sand fencing and beach nourishment)."*

**I believe that it would make more sense to me to call our attention to the fact that the shape of the dune, not only the aspect ratio as that might be a bit limiting indicator as the shape of a dune can vary very much, turning it very important to consider indicators that inform about the volume, in addition of course, to the elevation, as that is the key parameter that determines the impact regime, the impact can shift to overwash or inundation if a particular height is maintained over a certain width of the dune, but this is not informed by the aspect ratio. So, I would say that additional information other than only the aspect ratio would be needed to actually understand if a particular dune can cope with the impact of one or several storms if the shoreline is retreating (or if the dune is being eroded) the aspect ratio of the dune may also change, as they are all but very irregular features.**

We agree that there are many different metrics which can be used to describe the shape of the dune beyond the aspect ratio such as the dune face slope, dune height, or dune width. When we constructed the set of synthetic dune shapes, we set all the dunes to a similar volume so as to compare volume loss among the dunes as a consequence of their shape. To control for volume, we had to adjust both the height and width of the synthetic dunes (as described in the methods section of the manuscript). The aspect ratio metric takes into account both the height and the width of the dune and thus it describes the shape of the dunes we used due to the manner in which the synthetic profiles were constructed. Additionally, as a consequence of the manner in which we constructed the synthetic profiles, the dune face slope is directly related to the aspect ratio of the dunes and so all the results presented as a function of aspect ratio could be interpreted the same way as a function of the dune face slope given that these two parameters are not

independent of each other in this study. To address this concern, we have added the following table into the methods section as a complement to Figure 2 detailing shape parameters we use to more fully describe the dune shapes used in this study:

|  | Aspect Ratio (-) | Volume (m3/m) | Dune Height (m) | Dune Width (m) | Dune Slope (m/m) |
|---|---|---|---|---|---|
| **1.6X** | 0.25 | 52.75 | 5.68 | 49 | 0.15 |
| **1.4X** | 0.2 | 50.64 | 4.95 | 50 | 0.13 |
| **1.2X** | 0.15 | 51.76 | 4.29 | 53 | 0.12 |
| **1.0X** | 0.11 | 53.27 | 3.69 | 56 | 0.1 |
| **0.8X** | 0.07 | 53.45 | 2.9 | 60 | 0.08 |
| **0.6X** | 0.04 | 53.25 | 2.18 | 67 | 0.06 |
| **0.4X** | 0.02 | 53.29 | 1.47 | 82 | 0.04 |

For clarification we have also modified the caption for Figure 2 to highlight that the "#.#X" values refer to the percent increase (decrease) in dune height (width) relative to the base profile (1.0X) as described in the methods section of the manuscript to maintain a constant initial volume. We have also added the following lines to the section on synthetic dune profiles to address the limitations of these shapes (revised manuscript lines 132-134):

*"We modify the dunes to have a similar volume to more easily compare volume loss across different aspect ratios. For completeness in describing the dune shapes, the initial dune shape parameters (i.e., steepness, height, width) can be found in Table 1."*

**In this line, the authors state that the amount of dune erosion or the vulnerability of coastal dunes does not only depend on dune height but also on its width, as dunes can also erode by collision regimes, and thus, an aspect ratio that includes both dimensions should be used instead of only the elevation. In addition, they also state that dune erosion might be influenced by this aspect ratio. In general, I do agree with the hypothesis stated by the authors, however, I cannot fully agree with the approach used to support their statements and reinforce my concern regarding the originality of the contribution from this work. My main concern is linked to the experiments chosen by the authors, namely to the synthetic dunes used and the way the authors have decided to create the four different configurations, which ended up having dunes with different morphologies (symmetric, non-symmetrical and with changing front dune slopes) and different distances to the shoreline that cannot be easily compared. In fact, the results suggest other factors might be more important than the dune aspect; namely the beach width, which determines the level of impact of the storm over the dune**

We appreciate the reviewer's concern that the role of the beach width is critical in understanding how dunes erode during a storm and we have revised the paper (in the ways described on page one of this review) to clarify and strengthen this point. In addition to the changes made, we have kept the numerous instances in which we did refer to the importance of beach width in the original submitted version of the manuscript.

**I will try to synthesize my concerns focusing on some statements from the abstract of the manuscript and mainly linked to the results of the experiments or simulations.**

**The authors state in the abstract that "low aspect ratio (low and wide) dunes lose less volume than high aspect ratio (tall and narrow) dunes during longer storms, especially if they are fronted by a narrow beach". Regarding the first part of the sentence and looking at Figure 7, where the results from the simulations are presented, it is not so obvious this affirmation as low aspect ratio dunes only erode less when using the fixed dune toe configuration (narrowing the size of the beach).**

In response to comments from Reviewer 2 (described below), we ran new simulations with improved parameterizations for simulating erosion with XBeach (after Palmsten & Holman, 2012; Palmsten & Splinter, 2016; Splinter & Palmsten, 2012) to better replicate scarping and prevent XBeach from overestimating erosion (see response to Reviewer 2 comments for details). As such, our results are now more physically grounded and they also demonstrate a clearer distinction between how high aspect ratio dunes erode versus low aspect ratio dunes. Our updated Figure 7 (as described on page one in this response) shows a 10-40m$^3$/m difference in volume loss between high and low aspect ratio dunes, depending upon storm conditions. We have added text to the manuscript to better explain and quantify this revised result (Lines 213-233 in the revised manuscript):

*"Simulations with the dune toes aligned have profiles that all share the same beach width (and slope) for the different dune aspect ratios included in this study, thus decoupling aspect ratio from beach morphology. Simulations with the dune toes aligned showed that there was greater erosion for the high-aspect ratio dunes compared to the low-aspect ratio dunes (Figure 7A, B, C). The increased erosion was especially pronounced during low intensity storms where there was a ~10 m$^3$/m (~19%) difference in volume loss between the high and low-aspect ratio dunes. For the most intense storms, the difference in volume loss between the high and low-aspect ratio dunes was ≤10 m$^3$/m (~19%). As expected, increasing the duration of the storms led to an increase in the amount of overall erosion experienced, especially for high-aspect ratio dunes. While none of the dunes were completely inundated in our simulations, the dunes (all aspect ratios) lost a significant amount of sediment (>30m$^3$/m, >60%).*

*Although the tall/narrow dunes lose the greatest amount of sediment compared to the low/wide dunes, the dune toe experience less erosion regardless of the storm scenario. While the dune toe for the low/wide dune retreated up to 10m during the longest and most intense storms (Figure 7F), the dune toe for the tall/narrow dunes never retreated more than ~5m (Figure 7D) and actually moved seaward by up to ~12m (Figure 7F). For a given storm duration and intensity, the dunes of different aspect ratios are impacted by a comparable amount of wave energy (Figure 7G, H, I) and the beach morphology is the same for all simulations, so the style of erosion is purely being regulated by the morphology of the dune. High aspect ratio dunes are closer to the angle of repose such that they avalanche with sediment piling up at the dune toe. In contrast, low aspect ratio dunes lose more volume than the high aspect ratio dunes but the sediment is lost to the surf zone. For example, during a shorter (+10 hours) storm with low surge, dunes with an aspect ratio of 0.1 and 0.2 both lost equal amounts (~5m$^3$/m) of sediment (Figure 7A) and were impacted by an equal amount (~500Nm/m$^2$ ) of wave energy (Figure 7G)*

*during the storm. However, the sediment for the lower aspect ratio dune experienced dune toe erosion while the higher aspect ratio dune experienced dune toe progradation.”*

**The higher aspect ratios in this case are related to dunes with very steep seaward slopes resembling scarped dunes. As the authors state, and following Hesp 1988, these dunes can be more un-stable and have greater probabilities of crest collapse, is all the sediment from the col-lapsed dune removed by the waves? Having in mind the main principles of dune/beach erosion, the amount of volume eroded should not be very different or dependent on the dune shape, but on the volume of sand that needs to be eroded as it is a response to adjust the beach profile to more energetic waves.**

We agree that wave energy impacting the dune plays a critical role in describing how the dunes erode and also that sediment may be deposited at the dune toe or transported towards the surf zone depending upon the mode of erosion. In our simulations we are able to quantify the cumulative energy impacting the dune in each simulation. We have also quantified the change in dune toe position following comments from Reviewer 2 (a metric which relates to where sediment is deposited during the storm). We have added a description of these new metrics in the methods section (Lines 99-109, new text in *italics*):

“To track changes in dune and beach morphology throughout our simulations, the following morphometrics are calculated at every model time step: dune aspect ratio, dune volume, overwash volume, beach width*, dune toe erosion, and wave energy reaching the toe of the dune*. The dune aspect ratio was calculated as the height of the natural dune from $D_{high}$ to $D_{low}$ divided by the width of the dune from $D_{heel}$ to $D_{low}$ (Figure 1). The dune volume is calculated by integrating over the portion of the profile contained within the original cross-shore location of the dune ($D_{low}$ to $D_{heel}$) in the first-time step and above the $D_{low}$ elevation (0.59 m, NAVD88). The overwash volume is similarly calculated as the change in volume of the profile landward of the initial $D_{heel}$ position. Beach width is calculated as the cross-shore distance between MHW and $D_{low}$ at every time step. Given that pre-storm $D_{low}$ was held constant across all simulations, the beach slope is inversely proportional to the beach width in our simulations (i.e., beach slope decreases as beach width increases). *Dune toe erosion was measured as the final minus initial position of the dune toe. Wave energy is counted as the cumulative amount of wave energy at the dune toe throughout the simulation.”*

We have also included new metrics throughout the results section and in the new figures described above. From this analysis we find that, for a given storm condition (surge and duration), there is little difference in the energy impacting the low versus high aspect ratio dunes (when the toes are aligned) however the high aspect ratio dunes have a seaward shift in the dune toe position (sediment piling up at the base of the dune following scarping) while low aspect ratio dunes had a negative shift in dune toe position (sediment being carried offshore). This result would suggest that differences in volume loss are related to the dune aspect ratio as every other factor was controlled for in the toes-aligned simulations and there was minimal difference in the wave energy impacting the dunes. We have incorporated these findings into the results section for toes-aligned dunes, which can also be found on page **6** of this response.

Further, this comment by the reviewer is grounded in the assumption that the beach achieves an equilibrium profile during the course of the storms. Given that an equilibrium profile is not necessarily reached during the course of a storm, this would affect the volume of sediment lost and may be regulated by the aspect ratio. To explain this we have added the following sentences to the discussion section (revised manuscript lines 362-367):

*"Additionally, the dune aspect ratio may determine the rate by which the profile achieves equilibrium during a storm. If equilibrium is achieved than volume loss would be equal for both the low and high aspect ratio dunes. However, given that erosion is a time dependent process and is also dependent upon the levels of wave energy during the storm. It is possible that when the dune is scarped (a process which is more associated with the high aspect ratio dunes in our simulations) more sand is moved into the surf zone which will help the profile achieve equilibrium more quickly compared to a low profile dune where there is less sand moving into the surf zone."*

**It is not so obvious from these results that low ratio dunes erode less, as the authors change the shape of the dunes and their distance to the shore, and therefore, they cannot be easily compared any-more.**

The distance to the shore only varies between the different profiles in the crests-aligned and heels-aligned configurations. For these simulations, the beach width increases with the dune aspect ratio such that higher aspect ratio dunes are fronted by wider beaches (note that this is more pronounced in the heels-aligned setup as seen in Figures 2B and 2C). Because we only compare results within each configuration (i.e., Section 3.3.1 compares results from the toes-aligned simulations), and the dune and beach morphology are consistent across runs within each configuration, we believe the within configuration comparisons are valid. We only compare results *across* configurations to evaluate the effectiveness of the fenced dune in mitigating erosion (Section 3.2). However, with the changes to the structure of the paper described above we have made sure to more effectively and clearly distinguish the erosion attributable to the dune aspect ratioand the erosion attributable to differences in beach morphology.

**In this regard, I would imagine that a more synthetic dune would help and allow direct comparisons. By a more synthetic dune I imagine something that preserves the slopes (shape) changing the ratio (even though not so realistic, a cube would make it easier). In fact, when looking again at figure 7, the lower ratio dunes are eroding more for all cases but the toe fixed one, which is due to a change in the seaward slope of the dune and not merely to a change in the ratio, as those have very very gentle slopes that do not make easy to identify which is the actual location of the dune toe (or transition to the beach). So, from what is shown here, high ratio dunes are not "specially" losing more volume of sand when the beach is narrow as the authors state, but "only" when the beach is narrow, otherwise they almost don't lose sand.**

We agree with the reviewer's comment that different dune face slopes may lead to variability in how the dunes erode. We also made this point in the discussion section (lines 377-

379) where we pointed out that the higher aspect ratio dunes have dune face slopes that are closer to the angle of repose and thus may more readily experience avalanching/scarping.

**The latter results from the fact that the other configurations (fixed crest and heel) present wider beaches and so, they may enter the collision regime later, or not at all. The authors also suggest that this affirmation mostly applies to longer storms. However, from figure 7f, j, g, k, we see that the erosion increases over time also for low aspect dunes if the configuration of the dune is different, so, again, the problem of comparing dunes with different shapes and distance to the shore does not help interpreting the outcomes of this work, and to assess the role of the ratio.**

We agree that the results within the crests and heels-aligned configurations need to factor in the role of dune aspect ratio and beach width, which is why we have restructured this analysis to more clearly present erosion as a function of dune aspect ratio (Section 3.1 and Figure 7) and beach width (Section 3.3 and Figure 8). Additionally, we have added the following statements to clarify this distinction by explaining a) that the toes-aligned simulations control for beach width and thus we can attribute differences in erosion to differences in dune morphology and b) that by subtracting out the erosion from toes-aligned simulations from the crests- and heels-aligned simulations (as described above) we are able to isolate the effect of varying beach width between the dunes in these simulations. These additions are on Lines 209-214, 219-221, 264-268 in the revised manuscript, and shown in order below:

- *"Additionally, the four different dune configurations presented in our analysis (i.e., toes-aligned crest-aligned, heels-aligned, and fenced) allow us to isolate the amount of erosion attributable to the dune morphology (aspect ratio) and the amount of erosion attributable to the beach morphology (width and slope) by establishing a baseline level of dune erosion in the toes-aligned simulations before introducing variations in the beach width in the crests- and heels-aligned simulations and management interventions in the fenced simulations"*
- *"Simulations with the dune toes aligned have profiles that all share the same beach width (and slope) for all of the different dune aspect ratios included in this study; Because of this we are able to isolate the effect of dune aspect ratio on dune erosion from these results because the only difference between the simulations was the dune morphology."*
- *"To isolate the effect of beach width we subtract the amount of erosion from the dunes in the toes-aligned simulations (which control for beach width) from the amount of erosion in the crests- and heels-aligned simulations."*

**Also in the abstract, the authors affirm: "During more intense storms, low aspect ratio dunes experience greater erosion as they are more easily overtopped". I see again a problem when comparing dunes with different shapes and distances to the shoreline. The affirmation sounds totally logic to me as the amount of volume needed to erode depends on the magnitude of the storm, and as stated by previous authors: "one of the most significant factors affecting the magnitude of spatial and temporal change to a foredune during an**

**erosion event is the height of the mean water level during the storm(Davidson, Hesp and Miot da Silva, 2020 and references therein)", and in fact this is the main factor that the authors assess when changing the magnitude of the event, the water level. However, when looking again at figure 7 (namely 7i), high and low aspect ratios share maximum volume loses, and in fact the main difference with fig.7e, is the fact that a higher water level reaches the low aspect ratio dune crest more easily than low water levels. Conversely, low aspect ratio dunes for the configurations other than the fixed toe one, are more vulnerable because their toe is closer to the shoreline when compared to the high aspect dunes, which are far from the shore in the crest and heel fixed configurations, which in turn makes it difficult to compare volumes of erosion**

The changes described in our responses above better isolate the effect of varying beach width in our simulations. Specific to this comment, a new figure (Figure 10) now shows the difference in volume loss between the crests- and heels-aligned simulations and their equivalent toes-aligned simulations as a function of beach width. This shows a strong relationship between the amount of volume loss and the increase in beach width relative to the toes-aligned simulations. For example, we now show that for heels-aligned simulations with a ~68m wide beach there was a 15-30m$^3$/m decrease in volume loss (greater decrease in volume loss for longer storms) compared to the equivalent toes-aligned simulations which had an equally narrow beach for all dune shapes. In addition to the new figure, we have added an explanation for this relationship in lines 261-264 in the revised manuscript:

*"Combining results from all simulations (Figure 10), we observe a proportional relationship between pre-storm beach width and volume loss (relative to the equivalent toes-aligned simulation). We also find that the erosion mitigated by wider beaches is even greater under longer and stronger storms (Figure 10)."*

**Still in the abstract, the authors state that in managed scenarios (by managed dunes, they refer to those sites where a fenced dune is constructed seaward of the existing natural dune) a fenced dune effectively prevents the natural dune behind from experiencing volume loss. Again, this is what should be expected as having a fenced dune implies in their case that the shoreline is again seaward and distant from the toe of the natural dune, resembling a wide backshore. Yet, the authors mention that the volume loss can be reduced up to 50%, which is a number that depends on their experiments, as if they have built a larger fenced dune it would be greater, also if instead of using the toe fixed they had used the crest or heel fixed configurations, that % would increase. So, again, it is not clear the point/contribution of the authors as these are well grounded ideas/facts usually used for the managers to design the actions to take**

While our simulations show a reduction in volume loss of up to 50% with the new XBeach parameterization between the fenced and toes-aligned simulations, we fully expect the modeled volume loss to vary based on both the parameters used with XBeach and the configuration of the synthetic profiles. We have added this point in an explanation clarifying that the specific amount of volume loss related to how our dunes eroded are a function of XBeach parameterization but that we expect the trends and relationships identified in this work to hold

true in all instances given that any of these changes related to model setup would be applied uniformly to all simulations and would thus impact them all equally (Lines 209-211):

*"We note that while the specific amount of volume loss relating to dune erosion and wave energy are a function of model setup, the setup is applied uniformly to all simulations such that the trends will remain constant regardless of how the dune shapes are formulated and how the model parameters are set."*

Regarding the shoreline positioning in the fenced dune simulation, the shoreline is in the same location as it is in the toes-aligned simulations; The only difference is that part of the beach has been turned into the fenced dune. While this scenario may not be entirely realistic, it allows us to directly quantify the role that the fenced dune plays in mitigating dune erosion by allowing for a comparison between the fenced and toes-aligned simulation as the only difference between the two is the presence of the fenced dune. If the fenced dune were placed in the crests-aligned or heels-aligned simulations, the issue of the varying beach widths would still be present as the distance from the shoreline to the toe of the fenced dune would vary between the different dune profiles. While the role of sand fences on dune growth has been studied before (e.g., Anthony et al., 2007; Charbonneau & Wnek, 2016; Itzkin et al., 2020; Mendelssohn et al., 1991), a quantitative assessment of how much natural dune erosion is prevented during a storm by the presence of a fenced dune has, to our knowledge, not been previously addressed. To explain why we have configured the fenced simulations in this manner we have added the following sentence to the results section (revised manuscript lines 284-287):

*"We performed a suite of simulations using the same dune profiles as the dune toes-aligned scenarios but with a portion of the beach replaced by a fenced dune (Figure 1D). The placement of the fence on the beach allows us to compare with the non-fenced simulations without changing the beach width. By comparing the results from these simulations with those from the toes-aligned simulations (Figure 10) we are able to quantify the effectiveness of artificial dunes (formed via the emplacement of sand fences) under varying storm scenarios."*

We have also modified the following sentences in the discussion section (revised manuscript lines 393-394 and 416-420 respectively):

"The key dynamic in this case (regardless of actual storm duration), was that the fenced dune was sufficiently high to protect the natural dune until the peak of the storm had passed. The foredune behind a fenced dune is not impacted until the fenced dune is eroded away, making the aspect ratio of the foredune secondary to the morphology of the fenced dune in providing protection to back-barrier environments *(a taller fenced dune would offer even greater protection)."*

"The strong inverse relationship between beach width and dune erosion (Figures 9 and 10) suggests that regardless of the aspect ratio of a foredune, widening the beach can be sufficient for preventing overwash during most storms and will be a more effective strategy for increasing coastal protection than re-building the dune or installing sand fences; *although pairing sand fences with a wide beach via nourishment would offer the greatest overall reduction in natural dune volume loss."*

**In this line, the authors also state: "a wide beach offers the greatest protection from erosion in all circumstances regardless of dune morphology or storm characteristics". This is again, the expected result, so, are the authors trying to convince the readers that XBeach can simulate erosion? And yet, this is again something that many examples in the literature have demonstrated. Finally, and yet in the abstract, the authors end saying: "in maintaining wide beaches and dunes, the protection offered in the short-term must be considered against long-term detrimental effects of potentially limiting overwash fluxes, which are critical to maintaining island elevation as sea level rises". This idea is developed at the end of the discussion section. From what I understand, a nourished beach, if using the adequate sediments, could provide the needed sediments for the dune to cope with sea-level rise as that is the regular mode dunes grow vertically. The authors claim that maintaining large beaches would prevent rollover and thus, the natural adaptation of the barrier to sea-level rise. Here, I cannot agree with the authors because barriers, if sediment input is enough (so they maintain their beach, even if artificially) can adapt by adapting the vertical elevation of the dune feed from the abundant sediments of the beach, and not only through rollover, right? I see more the problem of being able of maintaining large beaches with the adequate sand and in terms of costs. In this context, if sea level rises, so will the nourished beach berm and thus the dune, unless winds are not efficient anymore, which seems not to be the case in the future. Therefore, I would recommend the authors to better explain what the actual message is that they try to highlight here and to state what is their opinion about the possible coupled adaptation of the dune in that context.**

At the end of the discussion section (Lines 477-485) we point out that the limited overwash flux in managed locations can have long-term implications for barrier island survival under sea level rise. Barrier island rollover under SLR and how it may be modulated by development is a process that is well-studied (see citations in the paper). Our work ties into this process because we show that wide beaches and higher aspect ratio dunes will be more resistant to erosion and therefore less frequently overwashed. Rollover is the cumulative effect of numerous overwash instances and previous studies show that reduction in overwash volumes can hasten island drowning (e.g., Magliocca et al., 2011; Rogers et al., 2015). If dunes (or structures) reduce the volume of overwash from reaching the island interior and back-barrier, island elevation decreases, and nourishment on the front side will not prevent drowning from the bay side. Further, while it is true that continuous nourishment can promote resilience to barrier drowning (i.e., Masselink and Lazarus, 2019), this is not a realistic long-term solution. Resource limitations, pointed out on line 500 and by the reviewer, will not allow for continuous nourishment indefinitely. We have revised the wording on these points at the end of the discussion to clarify (revised manuscript lines 421-430):

"*It is important to recognize that although management initiatives such as widening beaches and building dunes with particular aspect ratios can be effective at mitigating erosion, these actions may have effects that are undesirable in the long-term. For example, overwash facilitates barrier rollover—a process that is necessary if islands are to maintain subaerial exposure as sea level rises (e.g., Leatherman, 1979; Moore et al., 2010; Lorenzo-Trueba & Ashton, 2014; Rogers et al., 2015). Thus, constructing dune and beach systems that reduce the amount of overwash that would otherwise naturally occur may inhibit rollover, thereby*

*increasing the likelihood of eventual barrier drowning (e.g., Magliocca et al., 2011; Rogers et al., 2015). This lack of overwash is a concern even in the presence of expected ongoing beach nourishment because overwash-induced increases in island interior, and back-barrier elevations are necessary to prevent drowning from the backside and because it is not feasible to continue beach nourishment indefinitely along all developed barriers."*

**More specific comments: Regarding the introduction, I would expect here that the authors present the main factors controlling the magnitude of dune erosion by storms, as this will be the main point of discussion, also considering the scales of change, from a single storm to a series of storms or the time-scale of relevance for managing would be interesting. In relation to the methods, they mention that the effect of the configuration of the beach, namely the beach slope, will be also assessed, however, this is not thereafter clearly evaluated. In line 281, I would add here the importance of the width of the dune to the impact of storm groups also.**

We see the confusion here, perhaps introduced by our objectives which state that the third question we will address is "How does the morphology of the beach (i.e., width and slope) affect volumetric dune erosion independent of dune aspect ratio?"  As we explain in the methods section (2.1, Lines 94-96), because the dune toe elevation is the same (0.59m) for all the dunes considered, the beach width and beach slope (calculated between MHW and the dune toe) are directly proportional to one another. For this reason, we only consider beach width in our analyses because additional analyses on the beach slope would be redundant.  We address this concern by modifying the statement of our third objective (Lines 93-94) as follows (new text italicized): "How does the morphology of the beach (i.e., *width/slope*) affect volumetric dune erosion independent of dune aspect ratio?" We have also added a statement to the methods (Lines 106-108) explaining that beach slope and beach width are directly related in these simulations to further clarify this issues.

Additionally, we have included the following sentences to the introduction to clarify that the focus here is on a single storm and the specific metrics we analyze in the paper which can be found on page **3** in this review and lines 77-86 in the revised manuscript

We have also modified the following sentence in the discussion section to address the effects of multiple storms on the dunes as described in our comments on page 11-12 of this response.
* * *
The following figures can be found in the revised version of the manuscript referred to in the responses to the reviewers. Note that figure numbers refer to their placement in the manuscript.

[Figure]

**Figure 7:** Dune aspect ratio versus storm duration for simulations with toes-aligned (controls for beach width/slope and initial dune volume). Each column represents a different storm surge level (increasing left to right). The top row (A, B, C) shows the change in dune volume, the middle row (D, E, F) shows the change in dune toe position (negative values indicate landward erosion), and the bottom row (G, H, I) shows the cumulative wave energy impacting the dune.

[Figure]

**Figure 8:** Dune aspect ratio versus storm duration for simulations with crests-aligned. The values from the equivalent simulations with the dune toes aligned have been subtracted from the crests-aligned simulations to highlight the influence from the varying beach widths in the crests-aligned simulations. Each column represents a different storm surge level (increasing left to right). These values represent a comparison relative to the toes aligned simulation (where beach width is controlled for) such that the top row (A, B, C) shows the amount of volume loss prevented by the wider beach in these simulations, the middle row (D, E, F) shows the additional dune toe progradation induced by the wider beach width, and the bottom row (G, H, I) shows the reduction in wave energy reaching the dune due to the wider (and thus lower sloping) beach.

[Figure]

**Figure 9:** Dune aspect ratio versus storm duration for simulations with heels-aligned. The values from the equivalent simulations with the dune toes-aligned have been subtracted from the heels-aligned simulations in order to highlight the influence from the varying beach widths in the heels-aligned simulations. Each column represents a different storm surge level (increasing left to right). These values represent a comparison relative to the toes-aligned simulation (where beach width is controlled for) such that the top row (A, B, C) shows the amount of volume loss prevented by the wider beach in these simulations, the middle row (D, E, F) shows the increase in dune toe progradation induced by the wider beach width, and the bottom row (G, H, I) shows the reduction in wave energy reaching the dune due to the wider (and thus lower sloping) beach.

[Figure]

**Figure 10.** Volume loss from the crests-aligned and heels-aligned simulations minus volume loss from the equivalent toes-aligned scenarios versus the initial beach width for the crests- and heels-aligned simulations. The color corresponds to the dune aspect ratio and the shape corresponds to the surge level.

[Figure]

**Figure 11:** Dune aspect ratio versus storm duration for simulations with sand fences. The values from the equivalent simulations with the dune toes aligned have been subtracted from the fenced simulations in order to highlight the influence from the presence of the fenced dune seaward of the natural dune. Each column represents a different storm surge level (increasing left to right). These values represent a comparison relative to the toes-aligned simulation (where beach width is controlled for and there isn't a fenced dune) such that the top row (A, B, C) shows the amount of volume loss prevented by the fenced dune in these simulations, the middle row (D, E, F) shows the increase in dune toe progradation induced by the fenced dune, and the bottom row (G, H, I) shows the reduction in wave energy reaching the dune due to the fenced dune.

---

## Author Comment (AC2) · 14 Apr 2021

We would like to thank Reviewer 2 for their thoughtful comments and suggestions which are much appreciated and have helped to strengthen our manuscript. Note that all the revised figures referenced throughout the response can be found at the end of this document.

**Overall, I found the paper well written and quite succinct. From a point of view on was there a substantial increase in our fundamental understanding into dune erosion, I was less convinced. Not much in the paper surprised me or told me something I didn't know, but more reaffirmed my understanding/observations/past work. That's not to say that more couldn't be presented to improve the paper and provide further understanding that I think is unique to modelling work as you have high resolution results that you can interrogate more than you have presented here. By presenting more and digging more into the results I think you could better answer your three objectives above. For example: Your dune profiles were very different and in XBeach, erosion occurs if a cell is determined to have been 'wet' so since your higher aspect ratio dunes had more sand closer to the dune toe, they would expect to have more erosion volumes by the nature of the model and not necessarily by a physical meaning. XBeach dune erosion is purely ad-hoc. If a cell is wet, it compares it to your wet slope and erodes it if it's above this critical value. Realignment can also take place if dryslp is exceeded. None of this is really based on physics of dune erosion. Dunes hold much larger scarps under active erosion (See Palmsten and Holman paper for examples but many others as well including work by Erikson and Hanson -> dune notching paper , Larson Erikson and Hanson (2004) and all the work on dune impact models (Overton et al) all show this). The sand is typically (from my experience using XBeach) also immediately moved offshore (to keep the wetslp low) so the feedback mechanisms we'd see in real erosion are not there where slumped sand protects the dune toe. The model has limitations and I can accept those but I think you need to acknowledge them a bit more here and realize what we can (and cannot) learn from these results. Consider the very different dune aspect ratios you are considering and the distribution of sand in the cross-shore, it would be good to see dune toe recession presented as well as you refer to volumes (which I also think are needed) but when you align toe,**
**heal, center, and with each of the aspect ratios you change the distribution of the volume in the dune. So small events will erode a lot when the toe is aligned because there is a lot of sand up close, but dx (dune toe erosion) might be similar and this is a key variable of interest to engineers/managers. The model is a grid so you are 'eating away' at the dune 1 grid point at a time as a function of the predicted TWL. Default dry slopes in XBeach are also quite flat compared to what would be capable in active dune erosion (see for example lab studies of Palmsten and Holman 2012,https://www.sciencedirect.com/science/article/pii/S0378383911001633; Palmsten and Holman 2011, https://agupubs.onlinelibrary.wiley.com/doi/full/10.1029/2011JC007083;Palmsten and Splinter 2016, https://www.sciencedirect.com/science/article/pii/S037838391600017X- this latter one explicitly looked at XBeach and my memory is that to match the lab data they used dryslp almost 4x the default value to allow for near vertical scarping)**

Reviewer 2 suggested that we conduct additional analyses to better understand changes to the dune and beach morphology as a function of the dune aspect ratio. Following these

suggestions, as well as those of Reviewer 1, we now include wave energy (directly outputted from XBeach) in our revised paper because it is related to impact hours (which were suggested by Reviewer 2), but provides more insight into the amount of erosion found in our simulations. We have also included analyses for the change in dune toe position as a function of dune aspect ratio and storm duration, as suggested by the reviewer. We do not include changes in beach width (suggested by reviewer 2) because it is independent of dune toe position. However (following our response to Reviewer 1), we have more clearly isolated the role that varying beach widths play in mitigating dune erosion (see page one of this response for details). Each of the analyses presented in the revised paper have been conducted using updated and improved parameterizations to XBeach following comments and suggestions by Reviewer 2. Details on this are provided in the responses below. Because these suggested changes are referenced throughout the review, we list the changes we have made to the paper in response to this review below:

- Updated the methods section (Lines 99-109) to describe the new metrics that we added to our analysis
- Updated the results section to include analyses of changes in the dune toe position (suggested by reviewer 2) and wave energy reaching the dune.
- Updated Figure 7 to only show results from the toes-aligned simulations where the different storm surge scenarios are presented in each column (rather than each row) and the rows each show a different metric as a function of dune aspect ratio and storm duration. In the top row we show volume loss, in the middle row we show change in the dune toe position, and in the bottom row we show the cumulative wave energy impacting the dune. Additionally, we have removed Figures and 9 from the submitted manuscript and replaced them with ones showing the same metrics but for the crests-aligned and heels-aligned simulations. Figure 11 shows this analysis for the fenced simulations.

These new analyses allow us to better describe not just the amount of erosion experienced by the dunes in our simulations but also the manner in which they were eroded (i.e., sediment piling up at the base of the dune via scarping versus sediment being transported offshore), adding greater depth and context to our analyses.

Here the reviewer also suggests useful papers that have indicated different XBeach parameter values, which may be more appropriate than the values we used with the simulations presented in the original manuscript. We re-ran the simulations with the new parameterization, described below and, which qualitatively confirm our original results but with some quantitative differences. The default values for wetslp and dryslp in XBeach are 0.3 and 1.0 respectively. The wetslp value we used is equivalent to that used by Palmsten and Splinter (2016) but with a dryslp of 4.0 instead of the default 1.0. To improve the model results and simulate better erosion physics with XBeach we re-ran the simulations using an improved setup with parameter values updated from those published by Splinter and Palmsten (2012) and Palmsten and Splinter (2016) The following values have either been changed from a previous non-default value or have been set from their default value:

- Changed *eps* from 0.05 to 0.1
- Changed *facSK* from 0.30 to 0.15

- Changed *dryslp* from 1.0 to 4.0
- Set *hswitch* to 0.10
- Set *hmin* to 0.01

We have added the following statement to the methods section detailing the changes we made to the parameterization and some of XBeach's limitations as detailed by the reviewer (revised manuscript lines 180-187):

*"We used the XBeach (Roelvink et al., 2009) model to simulate the effects of the synthetic storms described in Section 2.2 on the profiles described in Section 2.1. We ran XBeach (version 1.23.5465) in 1D-hydrostatic mode with the break parameter set to roelvink_daly and the gamma parameter set to 0.52 to better capture the effect of swash processes on the reflective beach profiles (Roelvink et al., 2018) we also adjusted parameters related to wave breaking and dry sediment transport in order to more realistically simulate dune erosion processes given the tendency of XBeach to overestimate erosion with default settings (Palmsten and Holman, 2011, 2012; Palmsten and Splinter, 2016; Splinter and Palmsten, 2012). XBeach erodes the profile by comparing the slopes to the dryslp (if a cell is dry) parameter or wetslp (if a cell is wet) to determine how much erosion should occur to maintain these values. Palmsten and Holman (2011, 2012) show that wet sand can sustain much steeper scarps than dry sand. By using a particularly high value for the dry slope (dryslp = 4), we allow the dunes to maintain much steeper, and more realistic, scarps during the storms (Palmsten and Splinter, 2016). This realism allow us to better understand how the dune is eroding under collision when it is actively scarping during the storm by comparing dune toe migration to dune volume loss. A full listing of non-default parameters can be found in Table 2."*

**Can you also answer your objectives in terms of dune toe recession (as well as volume)to get a deeper understanding/picture of how dune aspect ratio effects overall erosion. One would expect that perhaps that higher aspect ratios might also have less dune toe recession as more sand is dumped onto the beach and may offer protection. I would also like to see plots of beach width change over the storm. This is some-thing you say is quite important to your results – wider beaches offer more protection. Something that other researchers have also shown to be quite important (eg. Plant and Stockdon, 2012. Probabilistic prediction of barrier-island response to hurricanes https://agupubs.onlinelibrary.wiley.com/doi/full/10.1029/2011JF002326;Beuzen et al.2019.Controls of Variability in Berm and Dune Storm Erosionhttps://agupubs.onlinelibrary.wiley.com/doi/abs/10.1029/2019JF005184) Beach width(or safe corridor width) is also a key parameter that engineers/managers are wanting. How does your beach width over a storm impact on the erosion – does it need to be completely removed or only a certain percent for the dunes are vulnerable. I think if you could present your results looking at multiple parameters (volume, dune toe retreat, beach width change, dune impact hours) then the reader would get a much richer understanding of the impacts these changes to dune aspect ratio/beachwidth/storm duration had on the study. Volumes themselves only tell a small part of the story.**

The reviewer suggests considering a number of other response in our simulations to further understand how dunes are eroding. For details regarding the changes stemming from this

comment please refer to the top of our response to Reviewer 2. We have included dune toe position change and cumulative wave energy (related to impact hours) to our analysis to better understand changes in dune sand volume. We considered change in beach width but found it similar to change in dune toe position so we did not include this variable, although in our re-structured manuscript (see response to reviewer 1) we have included a figure (Figure 10) that demonstrates more clearly how the beach width and dune volume loss are related regardless of dune configuration.

**Other Scientific Aspects to be considered: L35: "Considering that wave runup is most likely to impact the dune face (i.e., collision; Sallenger, 2000), which is more likely to affect the width of the dune rather than the height, is the most temporally common impact regime during a storm (Brodie et al., 2019; Stockdon et al., 2007), the width of the dune is an important predictor of how much erosion a dune might experience during a storm." I find this sentence really hard to read. Consider revising. As well, width won't be a predictor so much of the amount of erosion I would think, but of the erosive vulnerability of the dune itself. This paper might be of interest to you as it looks at both dune characteristics (height/width) and beach width in terms of erosion and flooding risks in storms: Leaman et al. (preprint, under review in Coastal Eng). A Storm Hazard Matrix combining coastal flooding and beach erosion. https://eartharxiv.org/repository/view/1753/**

Regarding the suggestion to line 35: To clarify this statement and address the role of dune width we have changed these sentences in the introduction to (revised manuscript lines 39-41):

*"Considering that wave runup is most likely to impact the dune face, collision (Sallenger, 2000) – which is more likely to impact the width of the dune rather than the height – is the most common impact regime during a storm (Brodie et al., 2019; Stockdon et al., 2007) and thus the width of the dune is likely to be an important predictor of how vulnerable the dune is to erosion during a storm (i.e., Leaman et al., 2020)."*

**L171: I am a bit concerned about leaving all other XBeach parameters as default as many studies have shown this isn't appropriate outside of the highly dissipative beaches for which the model was originally designed (along the Dutch coast). Leaving all other parameters as default has implications between overwash and collision regime erosion estimates as noted by previous researchers such as Passeri et al. and Simmons et al.. Not accounting for these processes will impact on your results. Why weren't these considered?, even is the cases were limited where overwash did occur? Others have also shown sensitivity of the erosion to parameters. Eg references below(note this isn't a complete list, just ones I could think of off-hand). Passeri et al. The influence of bed friction variability due to land cover on storm-driven barrier island morphodynamics https://www.sciencedirect.com/science/article/pii/S0378383917301114Simmons et al.Calibrating and assessing uncertainty in coastal numericalmodels https://www.sciencedirect.com/science/article/pii/S0378383916303234#f0030C4 Splinter and Palmsten. Modeling dune response to an East Coast Lowhttps://www.sciencedirect.com/science/article/pii/S0025322712002034**

We agree that XBeach cannot appropriately simulate behavior on reflective beaches in its default state. To account for this, we used model parameters values published in Roelvink et al. (2018) who found good agreement with field data from Duck, NC by setting the breaker formulation to "roelvink_daly" and the value of gamma to 0.52 for simulations. Additionally, our revised manuscript includes results with simulations from the updated parameterization described above (Lines 40-42, Table 1) to more appropriately simulate dune erosion. The reduction in erosion using these new parameterizations has eliminated instances of overwash from our simulations, which is consistent with the lack of overwash in our study site during the survey period of 2016-2020, so tuning the model for collision appears to be appropriate in this case.

**L183: "or when dunes are located closer to the shoreline (represented by the dune toes-aligned scenarios; Figure 7)." I am a bit confused by this as the effect of beach width would be shown not when the dune toes were aligned (and all beaches had the same beach width) but instead when the dunes were aligned at their crest or heel, which then changes their beach width. Ideally you should be comparing the cases for the same dune aspect ratio at these three positions to determine if effect of beach width. And this is repeated for each of the dune aspect ratios. This would be an interesting thing to see in my opinion (same dune aspect ratio plotted for the 3 positions within your dune toe, heal, crest align) to see how BW effects erosion for the same dune. Wider beaches offer a big buffer of sand that must be eroded before the wave action can get to the dune and frictional damping of the runup would also occur, lessening the probability of a dune experiencing wave impacts. Looking at dune impact hours could be interesting and provide some good insight here.**

In the revised version of the manuscript we have addressed this concern (brought up by both Reviewer 1 and Reviewer 2) by including a figure (Figure 10) that directly compares the beach widths for the crests-aligned and heels-aligned simulations to the amount of volume loss between the crests-aligned and heels-aligned simulations with their corresponding toes-aligned simulations. The toes-aligned simulations are not included in this analysis because the beach width is the same for all the profiles such that those simulations would plot as vertical line for all toes-aligned simulations. This allows us to isolate the role of the beach width and demonstrate how much erosion is prevented for a given beach width regardless of the dune configuration. Additionally, we have included wave energy into our analysis throughout the results section to consider the amount of wave action reaching the dune (more impact hours, the metric suggested by Reviewer 2, leads to more cumulative wave energy reaching the dune).

**L185: "situated farther from the shoreline (dune heels-aligned)" as above, I don't see how having the dune heels aligned also indicates they are further from the shorelines as each of these cases would have a different beach width"**

We agree and removed the parenthetical "(dune heels-aligned)" to make this sentence easier to understand and more accurate. The dunes that were farther from the shoreline experienced less erosion than those that were closer to the shoreline; The dunes fronted by the widest beaches are found in the heels-aligned configuration as a consequence of how the synthetic profiles were configured (Figure 2).

**L241: "Additionally, the sensitivity of the dune to decreases in storm duration was inversely proportional to the beach width such that dunes fronted by wide beaches were noticeably less sensitive to increases in storm duration than dunes fronted by narrow beaches (Figure 9)." – It would be great to see figures that show beach width change over the storm.**

Figure 10 in the revised manuscript shows the reduction in volume loss between the crests- and heels-aligned simulations and their equivalent toes-aligned simulations as a function of pre-storm beach width and for different storm durations. The restructured manuscript no longer includes this sentence or paragraph but in our revisions to the paper (Lines 265-310) we more clearly isolate and analyze the relationship between beach width change and dune volume loss.

**Specific Minor Editorial Comments: L75: 'aspect ratio' is repeated twice**

**L91: replace 'Dtoe' with 'Dlow' to match figure 1 and to remove confusion as I believe that Dlow=Dtoe.**

**L95: "Given that Dlow was held constant across all simulations" I think should be "Given that prestorm Dlow was held constant across all simulations".**

**L147: remove '.' in 'approximately.'**

We appreciate that Reviewer 2 also pointed out some grammatical and punctuation errors, which we have addressed in the revised version of the manuscript.

**Overall, I think the paper could be improved to provide a fuller understanding of the complexities of dune erosion and how dune aspect ratio, beach width and storm duration/intensity impact on the model results. I have provided a number of example references to consider, but I'd like to acknowledge here that these are limited to what I could recall off hand rather than providing a complete list of relevant resources. Please consider these as examples and you might find more suitable ones within these papers as well**

Thank you very much for the references and kind comments of our paper.
* * *
The following figures can be found in the revised version of the manuscript referred to in the responses to the reviewers. Note that figure numbers refer to their placement in the manuscript.

[Figure]

**Figure 7:** Dune aspect ratio versus storm duration for simulations with toes-aligned (controls for beach width/slope and initial dune volume). Each column represents a different storm surge level (increasing left to right). The top row (A, B, C) shows the change in dune volume, the middle row (D, E, F) shows the change in dune toe position (negative values indicate landward erosion), and the bottom row (G, H, I) shows the cumulative wave energy impacting the dune.

[Figure]

**Figure 8:** Dune aspect ratio versus storm duration for simulations with crests-aligned. The values from the equivalent simulations with the dune toes aligned have been subtracted from the crests-aligned simulations to highlight the influence from the varying beach widths in the crests-aligned simulations. Each column represents a different storm surge level (increasing left to right). These values represent a comparison relative to the toes aligned simulation (where beach width is controlled for) such that the top row (A, B, C) shows the amount of volume loss prevented by the wider beach in these simulations, the middle row (D, E, F) shows the additional dune toe progradation induced by the wider beach width, and the bottom row (G, H, I) shows the reduction in wave energy reaching the dune due to the wider (and thus lower sloping) beach.

[Figure]

**Figure 9:** Dune aspect ratio versus storm duration for simulations with heels-aligned. The values from the equivalent simulations with the dune toes-aligned have been subtracted from the heels-aligned simulations in order to highlight the influence from the varying beach widths in the heels-aligned simulations. Each column represents a different storm surge level (increasing left to right). These values represent a comparison relative to the toes-aligned simulation (where beach width is controlled for) such that the top row (A, B, C) shows the amount of volume loss prevented by the wider beach in these simulations, the middle row (D, E, F) shows the increase in dune toe progradation induced by the wider beach width, and the bottom row (G, H, I) shows the reduction in wave energy reaching the dune due to the wider (and thus lower sloping) beach.

[Figure]

**Figure 10.** Volume loss from the crests-aligned and heels-aligned simulations minus volume loss from the equivalent toes-aligned scenarios versus the initial beach width for the crests- and heels-aligned simulations. The color corresponds to the dune aspect ratio and the shape corresponds to the surge level.

[Figure]

**Figure 11:** Dune aspect ratio versus storm duration for simulations with sand fences. The values from the equivalent simulations with the dune toes aligned have been subtracted from the fenced simulations in order to highlight the influence from the presence of the fenced dune seaward of the natural dune. Each column represents a different storm surge level (increasing left to right). These values represent a comparison relative to the toes-aligned simulation (where beach width is controlled for and there isn't a fenced dune) such that the top row (A, B, C) shows the amount of volume loss prevented by the fenced dune in these simulations, the middle row (D, E, F) shows the increase in dune toe progradation induced by the fenced dune, and the bottom row (G, H, I) shows the reduction in wave energy reaching the dune due to the fenced dune.

---

## Author Comment (AC3) · 14 Apr 2021

We thank Reviewer 3 for their thoughtful comments and suggestions and would also like to wish them a speedy recovery.

Note that all revised figures referred to throughout the response can be found at the end of this document.

**Dear Authors,**

**Thank you for your patience awaiting the reviews of your manuscript. The third referee suffered an arm injury and was unable to write up a full review report but did send me via direct email a number of general comments and suggestions.**

**The first two review reports both identify some major limitations of the work presented here. Of key concern are the lack of novelty of the findings and inconsistencies be-tween the objectives, simulation scenarios, and conclusions. These concerns are shared by the third referee, whose comments I can paraphrase here as: /- Conclusions are stretched, and the impact of beach width on dune erosion is already well-known. Itis also obvious that dune erosion is less when an artificial dune is placed in front; the% reductions found are purely a consequence of the arbitrary geometry of the artificial dune.**

This concern has been shared among the reviewers and we understand that further work was needed to clarify the conclusions and analysis presented in this paper. Reviewer 1, in particular, shared this concern and stemming from their comments we have restructured the paper to better demonstrate the relative roles of dune aspect ratio and beach morphology in determining how dunes erode under storms of varying intensities and duration. To explain how we have revised the manuscript, we present an excerpt from our response to Reviewer 1 below, which outlines the changes which have been implemented:

We have reframed our presentation, restructuring the results section and reorganizing the discussion section to highlight that: 1) We isolate the effects of dune aspect ratio on dune erosion through analysis of our toes-aligned simulations (in which beach width is held constant). These scenarios allow us to also examine the effects of storm duration and storm surge level on dune erosion as a function of dune aspect ratio. 2) We then isolate the effects of beach width on dune erosion by subtracting the effect of dune aspect ratio on dune erosion as determined from the analogous toes-aligned simulations, and 3) Our fenced aligned simulations then provide additional insights into how dune erosion is influenced by emplacement of sand fences as a function of storm duration and surge level. We have also proposed to change our title from "The influence of dune aspect ratio, beach width and storm characteristics on dune erosion for managed and unmanaged beaches." "The relative influence of dune aspect ratio and beach width on dune erosion as a function of storm duration and surge level." For more details, please see page one of this response for the complete summary of changes in this regard.

 **/- Systematic exploration of dune erosion is a good idea, but the set of scenarios is not optimal and there is little interrogation of the details of the erosion process itself.**

To further analyze the erosion processes occurring in our simulations we have performed additional analyses investigating changes in dune toe position, beach width, and wave energy impacting the dune. Results from these analyses have been incorporated into our revised manuscript and can also be found in our response to reviewer 2.

**/- There is thus more to be found in analysis of the simulations, perhaps also regarding the parametrisations in Xbeach. Figure 7 shows some interesting features that are not fully discussed.**

This comment was also shared by Reviewer 2 and we repeat our response here:

Here the reviewer also suggests useful papers that have indicated different XBeach parameter values, which may be more appropriate than the values we used with the simulations presented in the original manuscript. We re-ran the simulations with the new parameterization, described below and, which qualitatively confirm our original results but with some quantitative differences. The default values for wetslp and dryslp in XBeach are 0.3 and 1.0 respectively. The wetslp value we used is equivalent to that used by Palmsten and Splinter (2016) but with a dryslp of 4.0 instead of the default 1.0. To improve the model results and simulate better erosion physics with XBeach we re-ran the simulations using an improved setup with parameter values updated from those published by Splinter and Palmsten (2012) and Palmsten and Splinter (2016) The following values have either been changed from a previous non-default value or have been set from their default value:

- Changed *eps* from 0.05 to 0.1
- Changed *facSK* from 0.30 to 0.15
- Changed *dryslp* from 1.0 to 4.0
- Set *hswitch* to 0.10
- Set *hmin* to 0.01

We have added the following statement to the methods section detailing the changes we made to the parameterization and some of XBeach's limitations as detailed by the reviewer (revised manuscript lines 180-187):

*"We used the XBeach (Roelvink et al., 2009) model to simulate the effects of the synthetic storms described in Section 2.2 on the profiles described in Section 2.1. We ran XBeach (version 1.23.5465) in 1D-hydrostatic mode with the break parameter set to roelvink_daly and the gamma parameter set to 0.52 to better capture the effect of swash processes on the reflective beach profiles (Roelvink et al., 2018) we also adjusted parameters related to wave breaking and dry sediment transport in order to more realistically simulate dune erosion processes given the tendency of XBeach to overestimate erosion with default settings (Palmsten and Holman, 2011, 2012; Palmsten and Splinter, 2016; Splinter and Palmsten, 2012). XBeach erodes the profile by comparing the slopes to the dryslp (if a cell is dry) parameter or wetslp (if a cell is wet) to determine how much erosion should occur to maintain these values. Palmsten and Holman (2011, 2012) show that wet sand can sustain much steeper scarps than dry sand. By using a particularly high value for the dry slope (dryslp = 4), we allow the dunes to maintain much steeper, and more realistic, scarps during the storms (Palmsten and Splinter, 2016). This realism allow us to better understand how the dune is eroding under collision when it is actively*

*scarping during the storm by comparing dune toe migration to dune volume loss. A full listing of non-default parameters can be found in Table 2."*

**The 'managed dune' set-up is rather limited, it should include modifications of the dune toe such as marram planting, sand ramps, and sand fencing, interventions that aim to widen an existing/eroded dune again.**

The managed setup is meant only to represent the effect of a dune system that has an established fenced dune. The impetus for this project arose from an exploration on the morphologic evolution of fenced versus natural dune systems in the Outer Banks (Itzkin et al., 2020). While we do not simulate other management interventions that also effectively widen the dune, our results can reasonably apply to the interventions mentioned by Reviewer 3 given that they would result in a decrease in the dune aspect ratio while maintaining the pre-intervention dune elevation. We have addressed this issue in the paper by including the following (revised manuscript lines 399-405):

*"While we do not explicitly simulate other interventions that would widen the dune (e.g., dune grass planting, sand ramps), these management strategies effectively widen the dune without adding elevation, thus leading to a lower dune aspect ratio than the pre-management condition. In this case, the results presented for sand fences likely apply to these situations as well. In all cases, if dune management actions are not paired with beach nourishment (i.e., Itzkin et al., 2020) then the wider dune will likely come at the cost of a slightly narrower beach. The amount of erosion during storms will likely decrease because of the lower aspect ratio (Figure 7), but the potential decrease in erosion will likely be offset by the erosion arising from the narrower beach (Figure 10)."*
* * *
The following figures can be found in the revised version of the manuscript referred to in the responses to the reviewers. Note that figure numbers refer to their placement in the manuscript.

[Figure]

**Figure 7:** Dune aspect ratio versus storm duration for simulations with toes-aligned (controls for beach width/slope and initial dune volume). Each column represents a different storm surge level (increasing left to right). The top row (A, B, C) shows the change in dune volume, the middle row (D, E, F) shows the change in dune toe position (negative values indicate landward erosion), and the bottom row (G, H, I) shows the cumulative wave energy impacting the dune.

[Figure]

**Figure 8:** Dune aspect ratio versus storm duration for simulations with crests-aligned. The values from the equivalent simulations with the dune toes aligned have been subtracted from the crests-aligned simulations to highlight the influence from the varying beach widths in the crests-aligned simulations. Each column represents a different storm surge level (increasing left to right). These values represent a comparison relative to the toes aligned simulation (where beach width is controlled for) such that the top row (A, B, C) shows the amount of volume loss prevented by the wider beach in these simulations, the middle row (D, E, F) shows the additional dune toe progradation induced by the wider beach width, and the bottom row (G, H, I) shows the reduction in wave energy reaching the dune due to the wider (and thus lower sloping) beach.

[Figure]

**Figure 9:** Dune aspect ratio versus storm duration for simulations with heels-aligned. The values from the equivalent simulations with the dune toes-aligned have been subtracted from the heels-aligned simulations in order to highlight the influence from the varying beach widths in the heels-aligned simulations. Each column represents a different storm surge level (increasing left to right). These values represent a comparison relative to the toes-aligned simulation (where beach width is controlled for) such that the top row (A, B, C) shows the amount of volume loss prevented by the wider beach in these simulations, the middle row (D, E, F) shows the increase in dune toe progradation induced by the wider beach width, and the bottom row (G, H, I) shows the reduction in wave energy reaching the dune due to the wider (and thus lower sloping) beach.

[Figure]

**Figure 10.** Volume loss from the crests-aligned and heels-aligned simulations minus volume loss from the equivalent toes-aligned scenarios versus the initial beach width for the crests- and heels-aligned simulations. The color corresponds to the dune aspect ratio and the shape corresponds to the surge level.

[Figure]

**Figure 11:** Dune aspect ratio versus storm duration for simulations with sand fences. The values from the equivalent simulations with the dune toes aligned have been subtracted from the fenced simulations in order to highlight the influence from the presence of the fenced dune seaward of the natural dune. Each column represents a different storm surge level (increasing left to right). These values represent a comparison relative to the toes-aligned simulation (where beach width is controlled for and there isn't a fenced dune) such that the top row (A, B, C) shows the amount of volume loss prevented by the fenced dune in these simulations, the middle row (D, E, F) shows the increase in dune toe progradation induced by the fenced dune, and the bottom row (G, H, I) shows the reduction in wave energy reaching the dune due to the fenced dune.

---

## Author Response (AR2)

We thank the editor for his comments and Reviewer 1 for a very thorough reading and review of our revision. Their comments are much appreciated and have assisted us in improving the paper further. We also appreciate the important comment from Reviewer 2. We have revised our results to indicate that we are confirming the role of beach width in dune erosion. And, we have further clarified and demonstrated the role that fenced dunes play in mitigating dune erosion. Our responses to comments are listed below. Changes to the text are shown in italics.

**The new version of the manuscript submitted by Itzkin et al. shows a clear effort to accommodate the concerns raised by the reviewers in relation to the previous version, I appreciate this effort and I see that the authors have addressed some of those concerns in a very elegant and convincing manner. In this regard, I congratulate the authors for this improved version of their manuscript where they have been able to reorganize the same data in order to present and explain the observations from the modelling in a way that can be more easily transferable and comparable.**

**Despite the improvements made on the manuscript, I maintain some of my concerns, namely regarding the major contribution of this work, and new concerns related to the way some of the data is presented in this new version. I maintain my view that demonstrating that a wider beach protects more the dune should not be the major output or fact to demonstrate within a work, as I insist, that is the basis of mitigation practices involving beach nourishment. So, I insist that it should not be shown as a novelty fact, as we have all learned from before. I see that the authors prefer to maintain their point of view and insist that this fact has not been demonstrated before, then I ask the authors what principles the engineers have used so far in order to plan these measures as the dimension of the berm is a key element?**

**It is not easy to understand why the authors insist also in looking at the ratio to understand different erosion patterns and as the way they present it, mechanism. From their data, and previous knowledge that they also have, the only major factor that seem to influence the mechanism or pattern of dune erosion is the stoss slope. As they mention, very high dunes with scarps (similar to the high ratio in the toe-fixed profiles) could have the tendency to collapse, and then add a different mechanism into the dune dismantling when compared to the low ratio dunes. Other than this, I cannot see how the shape of the dune influences the erosion mechanism as the authors express.**

**The width of the beach determines the amount of volume eroded from the dune, but the author affirm that the width of the dune is also an important factor, I would rephrase this as what the latter determines is longevity of the dune to survive successive erosive events, as the dune will erode in the same manner but the wider the dune, the longer it will last under erosive processes.**

To address concerns from both the reviewer and the editor regarding the abstract of our manuscript, we have updated the abstract to better describe the changes to manuscript from the previous round of revisions. We have further revised it to explain that our results "confirm" that the beach width is the dominant control on dune erosion and that the width of the dune will determine how long the dune can persist under erosional conditions.

**One more very important point that raised doubts regarding the approach, is the way the dune changes in the dune are evaluated for the case of the crest- and heel-aligned profiles. The authors have opted for presenting the volume change of the latter relative to the**

**change for the toe-aligned in order to isolate the effect of the ratio and focus on the beach width. As a result, the variation in volume will be always positive because of the wider beach that the latter profiles present relative to the toe ones. I find this way of presenting the data all but intuitive, I believe that the authors can find a way to show the results in a manner that does not raise doubts as what one reads from the graphs is that the larger storm accumulated larger volumes of sand as the beach width or dune ratio increases. It is obvious that with a greater beach the volume of erosion from the dune will be always smaller than if you buffer zone is reduced, then the comparison among the volumes will be always positive (figures 9 and 10) but has a very different meaning from the results in figure 8. Then I suggest changing the name of the axis and thus of this estimate, as it is a delta volume but a relative one or in other words, the volume preserved in the dune because of the increase in the beach width. And the same applies to the dune toe.**

The positive values are a consequence of subtracting a large negative value (volume loss from the toes-aligned simulations) from a smaller negative value (volume loss from either the crests or heels aligned simulations). We explained this in lines 259-264:

To isolate the effect of beach width, we subtract the amount of dune erosion (i.e., volume, toe position change, and wave energy) that occurred in the toe-aligned simulations (which control for beach width) from the amount of erosion in the crest-and heel-aligned simulations. This calculation yields a positive number for volume change and dune toe migration, representing erosion that is prevented by the increase in beach width, and a negative value for wave energy representing additional wave dissipation provided by the beach for both the crest-and heel-aligned simulations.

To clarify this, we have revised the above statement as follows:

To isolate the effect of beach width, we subtract the amount of dune erosion (i.e., volume, toe position change, and wave energy) that occurred in the toe-aligned simulations (which control for beach width) from the amount of erosion in the crest-and heel-aligned simulations. This calculation yields a positive number for volume change and dune toe migration, representing *the volume of sediment preserved in the dune as a consequence of increasing the beach width*, and a negative value for wave energy representing additional wave dissipation provided by the beach for both the crest-and heel-aligned simulations.
* * *
**Below I will transfer some more specific comments from the reading of the manuscript that I hope can help the authors to further improve the manuscript.**
**From the abstract:**
**I believe the authors have not updated the abstract to incorporate the slightly different outcomes that they got within this new version. In this line, I would suggest the authors to adapt this new version to what is inside the manuscript, namely the authors state:**
**"We find that low aspect ratio (low and wide) dunes lose less volume than high aspect ratio (tall and narrow) dunes during longer storms, especially if they are fronted by a narrow beach."**
**This sentence was fitting the previous version but need adaptation as from my understanding the beach width is the major factor and then minor factor the ratio. Also,**

**for the particular case of the toe aligned and low surge case, the different in volume change is not so dramatically different as the authors try to express.**

Original sentence: We find that low aspect ratio (low and wide) dunes lose less volume than high aspect ratio (tall and narrow) dunes during longer storms, especially if they are fronted by a narrow beach.

Revised sentence: We find that low aspect ratio (low and wide) dunes lose less volume than high aspect ratio (tall and narrow) dunes during longer *and more intense* storms *when the beach width is controlled for*.

For the toes aligned simulations with low surge, there is still a difference in volume change across the different aspect ratios for each storm duration even if the gradient is not as strong as it was for the previous simulations. We do not claim any degree of a difference beyond the decreased erosion for the low aspect ratio dunes in these simulations.
* * *
**"During more intense storms, low aspect ratio dunes experience greater erosion as they are more easily overtopped than high aspect ratio dunes."**
**First, I cannot see how this affirmation is novel or bring something we did not know. Second, this affirmation tries to fit what is observed in figure 8C, where the very low dunes are easily eroded, however there is not a gradient in the sense that if the ratio increases, then erosion drops, but a jump in the dune response related to the magnitude of the surge. In fact, the affirmation refers only to dunes with ratios very close to zero while high ratio dunes AGAIN erode more during longer storms, in this regard, how is this different from the low or from the approach by Sallenger?**

We have removed this sentence given that the instances of overtopping observed in the original set of simulations were not observed in the new simulations, and this sentence becomes redundant with the sentence addressed in the above comment from the reviewer without the references to the aforementioned instances of overwash.
* * *
**Finally and also in the abstract, I was a bit disappointed to see that the authors maintained the fenced profiles. I wonder why they have not separated the effect of the fenced dune from the wider beach as well in this case? Because, the fenced profile does not only present a fenced dune but a wider beach. so, again, what are we actually looking at here? the combined effect of wider beach with the addition of a small dune, so additional volume of sand. The final result of a reduction in 50% might depend on the size of the fenced dune but also on the width of the backshore that you had to add to fit in the fenced dune. Your results also show that the increase of beach width may reduce up to 100% in dune erosion, therefore, why do you show the case of the fenced dune? I here would give this a second thought because of the message that these results are going to pass.**

Given that the editor has left it up to us to determine whether to keep the fenced simulations, we are opting to include them in the manuscript. We are keeping these simulations for two key reasons:

1. Sand fences are a common feature along managed coasts but their impact on the morphology of the dune system has been understudied. Demonstrating how the fenced-natural system erodes during different storm scenarios is a new finding. It is also interesting that the 50% reduction in erosion offered by the fenced dune was observed during the original and current set of simulations. To clarify that we can separate the role of the fenced dune from the role of the beach width, we have added the following sentences to the revised manuscript:

Lines 179-183: The series of fenced profiles is the same as the series synthetic natural dune profiles in which the $D_{low}$ position is aligned (Figure 3A) except that we added a gaussian curve on the seaward side of the dune to represent the presence of a typical fenced dune shape (Itzkin et al., 2020). *Similar to the simulations with $D_{low}$ aligned, our fenced dune simulations control for the morphology of the beach fronting the fenced-natural dune system (Figure 3D) and therefore isolate the role of the fenced dune in mitigating natural dune erosion.*

Which compliments the following sentence from results section 3.2:

Lines 336-338: By comparing the results from these simulations with those from the toe-aligned simulations (Figure 11) *we quantify how adding a fenced dune seaward of the natural dune affects dune erosion under the varying storm scenarios (while controlling for the effects of beach width).*

2. The section on fenced dunes is self-contained but helps to support the management implications highlighted throughout the discussion section.
The compounding role of the beach width and the fenced dune in preventing dune erosion is discussed throughout section 4.2 as well as in the following from the conclusion.  We have modified this text to further clarify:
Although modifying the dune aspect ratio *and/or constructing a fenced dune* does alter the amount of erosion experienced as storm characteristics vary, we find that the greatest protective service in all instances is offered by a wide beach; a finding that is *consistent with previous assertions* and also supported by our limited observations of dune erosion in the field.
       Our results indicate that a tall, wide foredune fronted by a fenced dune and a wide beach *offers the greatest protection from erosion.*

**Lines 30 and 31: The authors call our attention to the fact that long-term nourishment may have a negative effect on the capacity of the system to maintain its elevation. I believe that the authors are totally forgetting or dismissing the aeolian sediment transport, do you have any special reason? I agree with the authors that nourishing is not an eternal solution but because of its un-sustainability, but I do believe that the systems if they have positive budgets do not need to rollover through overwash to adapt the sea-level rise if they can also maintain it through aeolian transport. Of course the elevation of the berm will be controlled by the runup levels and then the sand only needs to be transferred inland by the wind.**
It's not clear what sentence this comment is referring to as we do not address nourishment in the abstract, however the concluding sentence of the abstract that talks about overwash and island elevation is consistent with the literature (Leatherman, 1979, 1983; Moore et al., 2010) stating that barrier islands must rollover on century timescales and beyond to maintain their subaerial

elevation. We agree that aeolian processes are part of building and maintaining island elevation, but we are not aware of any studies that suggest these processes are sufficient to prevent barrier islands from needing to migrate inland to respond to SLR.
* * *
**Line 60: "while the role that dune height…".**
**I disagree with the authors. I rather understand that the overwash potential has been widely explored and in this line the dune height appears as a passive parameter. I cannot find many works where the dune erosion depending on the storm is analysed as you are doing with the ratio. what i mean is that you seem to explore how the shape of the dune determines the degree of erosion, and not only the impact as other works (Sallenger) did.**
Original Sentence: While the role that dune height plays in determining storm impact has been well studied, the role played by dune width is less clear.
Revised Sentence: While the role that dune height plays in determining storm impact has been well studied, *it is less clear how dune shape (here, characterized as an aspect ratio) influences the style and magnitude of dune erosion.*
* * *
**Lines 68-69: "While vegetation zonation controls the positioning and height of the dune, the dominant plant species can influence overall dune shape", this is a bit confusing. Most works point to the sediment budget as the main factor controlling foredune height (Moore et al., 2016, Psuty).**
We understand where the confusion comes from in that earlier works do point to the sediment budget as a control on dune shape and appreciate the reviewer's reference to Moore et al., 2016. However, recent work has clarified and expanded previous understanding revealing complexities. First, one must separate the sediment supply to the beach (e.g., supply can be evidenced by progradation as the reviewer indicates) and sediment supply to the dune (flux from beach to dune). The current understanding is that sand flux to the dune determines how quickly dunes form whereas the distance from shore that dune-building vegetation can establish is important in controlling dune height because of the effects of topography on the wind field and thus the shear stress exerted on the beach (see Duran and Moore, 2013, for example). In Moore et al., 2016 the effect of progradation is not to increase sediment flux to the dune, but rather to change the distance from shoreline to the dune vegetation line, yielding dunes of different heights, depending on the interplay between seaward shoreline growth, lateral dune growth and relocation of the vegetation line. Since the sentence in question is consistent with the latest literature and it is beyond the scope of this paper to explain the complexities here, we have left this sentence as is, but we have added a phrase to the previous sentence to refer readers to Moore et al., 2016 in case there is interest in better understanding the dynamics we point to here.
Dune cross-shore position and dune height are controlled by the distance between the shoreline and the seaward limit of vegetation, with longer distances typically being associated with the formation of taller dunes (Durán and Moore, 2013; Hesp, 2002; *complexities occur on prograding shorelines, see Moore et al., 2016*) While vegetation zonation controls the positioning and height of dunes, the dominant plant species can influence overall dune shape (e.g., Biel et al., 2019; Hacker et al., 2012; Woodhouse et al., 1977; Zarnetske et al., 2010, 2012).

**Line 102-103: "Beach nourishment may also be used to widen the beach (and decrease its slope), limiting wave impacts to the dune and stimulating dune growth"**
**Here you are using what is widely known about the effect of beach width, then why this need to prove it again?**

These previous works do not directly assess the effect of beach width on how dunes erode. Van Puijenbroek et al. (2017) states that increasing the beach width stimulates dune growth. Ruggiero et al. (2001, 2004) addresses how changing the beach slope modifies the total water level and wave energy reaching the dune. Cohn et al. (2019) compares dune volume loss versus beach slope. The relationship between beach width has been explored conceptually (i.e., Burrough and Tebbens, 2008; Claudino-Sales et al., 2008; Silva et al., 2018; Itzkin et al., 2020; Davidson et al., 2020; Leaman et al., 2020), here we confirm this relationship quantitatively through the use of a numerical model. Future work could be done by running these simulations on profiles that control for beach slope while varying their widths. We have clarified this by stating that we are confirming the role of beach width in mitigating dune erosion as suggested by the editor. For example:

Lines 22-24: We then control for dune morphology to assess volume loss as a function of beach width and confirm that beach width exerts a significant influence on dune erosion

Lines 413-416: This result, combined with the results of the toe-aligned scenarios confirms that beach width is the primary control on the volume of sediment eroded from dunes during storms. Our results further suggest that dune width and dune height also affect the volume of sediment eroded by determining the "longevity" of the dune (thereby affecting the transition from avalanching to overwashing, e.g.) under erosive conditions.

Lines 472-475: Although modifying the dune aspect ratio and/or constructing a fenced dune does alter the amount of erosion that occurs as storm characteristics vary, we find that the greatest protective service in all instances is offered by a wide beach; *a finding that is consistent with previous assertions* and also supported by our limited observations of dune erosion in the field.
* * *
**Line 116: "the main goal…"**
**I believe the goal of the work is to assess dune erosion and you use a model to do so, but not all the way around.**
Thank you, yes. This is a much better way to express our goal.

 Original sentence: The main goal of the work presented here is to use a numerical model to assess how dunes erode during a single storm as a function of dune aspect ratio, beach width, and sand fence construction.

Revised sentence: The main goal of the work presented here is to assess how dunes erode during a single storm as a function of dune aspect ratio, beach width, and sand fence construction.
* * *
**Line 122: "the relative role of beach width in dune erosion processes requires further investigation (Davidson et al., 2020)."**
**This is not what the authors are stating, in fact the authors have not doubt and they present the beach width as a major factor preventing dune scarping, what the authors mention is that further research could be done on: "Surfzone–beach type controls on beach erosion and the degree of scarping. Observations and tests that record the difference in the magnitude or degree of scarping occurring on reflective versus intermediate versus dissipative beaches under the same storm and offshore swell conditions would be very**

**useful." Any reader can understand that this is quite different from the effect of beach width, which effect is well-known.**

We have removed the reference to Davidson et al., 2020 as we can see that the connection is tenuous. The revised text reads:

While previous studies of storm impacts have primarily focused on dune height (e.g., Long et al., 2014; Sallenger, 2000; Stockdon et al., 2007), recent studies suggest that dune width may also be a key predictor for how much dune erosion will be experienced (Leaman et al., 2020). Additionally, although beach width has been posited as a strong predictor of dune erosion (e.g., Burroughs and Tebbens, 2008; Claudino-Sales et al., 2008; Itzkin et al., 2020; Silva et al., 2018), we seek to quantify and understand the relative role of beach width in dune erosion processes.

**Line 136: "dune toe erosion"**
**Do you mean retreat or erosion?**

Revised for consistency with the rest of the paper.
Original sentence: Dune toe erosion was measured as the final minus initial cross-shore position of the dune toe.

Revised sentence: Dune toe *retreat* was measured as the final minus initial cross-shore position of the dune toe.

**Line 142: I would not include any mention to the beach slope. First because you are not using it in any of your plots and second because the beach slope computation should not include the backshore beach. any equation that you apply to estimate the runup etc. so, the processes to which the slope of the beach could have any effect are related to waves, and therefore they apply the foreshore slope or the combined foreshore and nearshore or only nearshore, but never the backshore because that is only important for the overwash potential, not for the runup, in fact you could induce to error because you may have a wide dry beach (as it is the case of your crest- and heel-aligned profiles) but the foreshore is the same and therefore the runup should not change.**

Original sentence: Simulations with the $D_{low}$ position aligned ensured that all scenarios share the same beach morphology, thereby controlling for the effects of beach slope on wave runup (i.e., Stockdon et al., 2006).

Revised sentence: Simulations with the $D_{low}$ position *control for the morphology of the beach fronting the dune to isolate the role of dune morphology on erosion*.

**From section 3.1 Erosion on synthetic Dunes:**
**Lines 258-261: "Foredunes erode under most simulated conditions, except when they have a high dune aspect ratio, are situated farther from the shoreline, or when the storm is of low intensity, in which case there is slight accretion at the dune toe due to wave processes (e.g., Cohn et al., 2019, Figure 8)."**
**If referring to fig. 8, please review this affirmation as I cannot see what you describe to see. How do you notice the wave processes? could you explain it using the graph? Because you mention this here but not in the subsections, so it is disconnected from the data. If you refer**

**to all your experiments, then you should refer to them or to cite all the figures, not only to fig.8. I cannot agree that high aspect ratio do not erode, at least from your results and here you should only refer to fig.8 experiments as otherwise you are looking at the effect of the beach width. So, please fix this as it might be the case that remained from the previous version.**

We agree with the reviewer that this sentence does not translate accurately from the previous version of this manuscript, and we appreciate the reviewer's astute observation of a change needed. To fix this oversight, we have revised this sentence to read: Foredunes erode under most simulated conditions (Figures 8, 9, 10, and 12), except when they are situated farther from the shoreline (Figure 11).
* * *
**Section 3.1.1:**
**Line 277: "..especially pronounced.."**
**I would rather say "clear" or I would try to lower the importance as it is not so pronounced. Also, for the intense storm and long duration the difces are not obvious anymore as well. What about the low ratio dunes which have the greatest losses?**

We have changed the phrase "especially pronounced" in this sentence to say "clear" following the reviewer's comment.

The small difference in erosion between the low and high aspect ratio dunes for the intense storm, long duration simulations is addressed in the following sentence, which we have revised for increased clarity: For the *longest and* most intense storms, the difference in volume loss between the high and low-aspect ratio dunes is $\leq 10$ m$^3$/m.
* * *
**Line 306: "In some cases, the tall/narrow dune toes prograde seaward by up to ~12m"**
**If this is related to avalanching, it should be more clearly stated. If that is the case, maybe I would give it a second thought and see if the dune actually eroded? Are you computing the avalanched dune sand as erosion? Could you explain this with more detail?**

The erosion (change in dune volume) is calculated as the change in the volume of sediment contained within the initial dune region as we explain in the methods (Lines 103-104): The dune volume is calculated by integrating over the portion of the profile contained within the original cross-shore location of the dune ($D_{low}$ to $D_{heel}$) in the first-time step and above the $D_{low}$ elevation (0.6 m, NAVD88). Given this, the dune can experience erosion (negative change in volume) while the toe prograde seawards when avalanching occurs.

We have revised the line as follows:
In some cases, the tall/narrow dune toes prograde seaward *likely via avalanching* by up to ~12m (Figure 8F).
* * *
**Section 3.1.2:**
**Lines 333-334: "... no appreciable increase in the amount of protection offered by the narrowest beaches as storm duration increases, …"**
**Is this needed? Why don't you also include what is left to be eroded from the beach to**

**exemplify this? If the beach was already eroded because it is narrow, how it could increase the amount of protection as the storm duration increases? This affirmation or perspective is missing the beach morphodynamics and therefore it seems rather pointless.**

We have removed this sentence as it is clearer to explain this through the results presented in Figure 11 and explained in the sentences that follow the one highlighted by the reviewer.
* * *
**Line 341: "…that wider beaches lead to a greater seaward migration of the dune toe." This again needs further explanation, why when you subtract the volumes then the change in volume relative to the toe-aligned profiles is volume loss prevented and the dune toe is interpreted as additional progradation? are not you just subtracting the retreat of the alternative experiments from the toe-aligned? If so, then is again distance prevented to migrate inland, but not progradation. Besides, how would you explain dune progradation? or is this related again to avalanching? But is it normal to have 30m (around double) progradation due to avalanching? Is this because the erosion is less intense? Or in other words, the energy reaching the base of the dune decreased.**

We have deleted the portion of the sentence highlighted by the reviewer as the first part of this sentence explains the effect seen here more clearly without it:

The final dune toe position is consistently farther seaward of the initial dune toe position for all dunes fronted by wider beaches than it is for the equivalent toe-aligned simulations and this effect was proportional to the beach width.

We have also added the following sentence for clarity (Lines 341-):

During the crest- (heels-) aligned simulations when these same dunes are fronted by wider (widest) beaches, *the post-storm dune toe location* of the high aspect ratio dune is *located* 15m (30m) *farther landward* compared to those observed in the toe-aligned simulations (where the toe of the high aspect ratio dune does not change) while the *post-storm dune* toe of the low aspect ratio dune is unchanged relative to the toe-aligned simulation (Figure 9F, Figure 10F). *The more seaward final dune toe position in the crests- (heels-) aligned simulations, compared to that in the toes-aligned simulations, is likely arising from avalanching of the dune as well as wave driven sediment transport to the beach/dune (e.g., Cohn et al., 2019).*
* * *
**Lines 342-343: "…while the toe of the lowest aspect ratio dune retreated by ~10m during the same storm during the toe-aligned simulations (Figure 7F)."**
**This makes sense as the beach is wider, but how do you explain that the same dune (low ratio) with a wider beach shows progradation? could you explain the process?**
**Also, I imagine you intended to refer to Figure 8, please fix.**

We have fixed the figure citation to Figure 8.

This sentence is in reference to the simulations presented with the toes-aligned, wherein the beach width is the same for all simulations. The differences are explained throughout section 3.1.1. We have added the following parenthetical here for clarity:

For example, the toe of the highest aspect ratio dune (*toes-aligned*) did not migrate during the longest storm with 1.5X surge while the toe of the lowest aspect ratio dune retreated by ~10m during the same storm during the toe-aligned simulations (Figure *8*F).
* * *
**Line 345: "progrades 15m (30m) landward compared"**
**What do you mean by landward progradation?**

This line was corrected while addressing the reviewer's comments for Line 341. Copied here:

During the crest- (heels-) aligned simulations when these same dunes are fronted by wider (widest) beaches, *the post-storm dune toe location* of the high aspect ratio dune is *located* 15m (30m) *farther landward* compared to those observed in the toe-aligned simulations (where the toe of the high aspect ratio dune does not change) while the *post-storm dune* toe of the low aspect ratio dune is unchanged relative to the toe-aligned simulation (Figure 9F, Figure 10F). *This change is likely driven through avalanching from the dune as well as wave driven sediment transport to the beach/dune (e.g., Cohn et al., 2019).*
* * *
**Lines 347-349: "Wave energy reaching the dune is reduced by up to 6000 Nm/m2 for the high aspect ratio dunes during the most intense storms with the widest beaches (heel-aligned; Figure 10I) while the energy impacting the dune is reduced by 1000 Nm/m2."**
**You are still mixing up ratios and beach widths. The energy reduction is because of the widest beach, so do not mention the ratio as it is not the factor controlling that, and then could you explain better the second part of the sentence in relation to the first part? Is something missing in the sentence?**

Revised to: Wave energy reaching the dune is reduced by up to 6000 $Nm/m^2$ during the most intense storms in simulations with the widest beaches compared to a reduction of 1000 $Nm/m^2$ when the dunes are fronted by a narrower beach (heels-aligned; Figure 10I).
* * *
**Section 3.2:**
**General comment: I still do not get the need of this simulations, also, as you were discussing the effect of beach width, could you explain how much different is this from enlarging the beach?? or how much extra-protection is due to the fenced dune and how much to the wider beach width?**

**Lines 358-359: "Additionally, the aspect ratio of the dune behind the fence plays a minimal role in influencing volume loss except in the case of the most intense storms,"**
**Is there a need to state this? it is too obvious that if the width of the beach is enlarged, and even you add a small dune in front, you would have to have a very long group of storms to reach the dune behind, i only see the point of this experiment in long-term simulations, not at the same temp scales as the previous as you have already shown that the beach width already prevents erosion**

The fenced dune simulations all have the same beach width, which we explain in the newly added lines referenced on page 3 in this document as well as Lines 285-288 within the results:

We performed a suite of simulations using the same dune profiles as the dune toe-aligned scenarios but with a portion of the beach replaced by a fenced dune (Figure 1D). By comparing the results from these simulations with those from the toe-aligned simulations (Figure 11) we quantify the effectiveness of artificial dunes (formed via the emplacement of sand fences) under varying storm scenarios while controlling for the effects of beach width.

These simulations show the erosion prevented purely because of the presence of the fenced dune. To the extent that we can compare the mitigation from the fenced dunes and increased beach width, this is performed in discussion section 4.2 where the management implications of this study are discussed (see also the bottom of Page 3 and top of Page 4 in this response).
* * *
**Lines 364-366: "… 21C). While the presence of a fenced dune prevents volume loss from the natural dune, there is little to no change (<10 m) in the dune toe position relative to the toe-aligned simulations where the fenced dune was not present (Figure 365 21D, E, F)." Please fix the reference to the figure to 12.**
**Could you explain this better, it is not clear what you are trying to express. How do you explain that if the dune is not eroded (you say "prevented volume loss") the position of the dune toe changes less than for the toe-aligned profiles without fenced dunes in front?**

We agree that this sentence is confusing and have determined that it is not necessary and so we have deleted it. This effect is more clearly articulated in existing lines 337-343:

We find that the fenced dunes prevent more volume loss (up to ~20 m$^3$/m) as the surge increases (Figure 12A, B, C) however, for any given surge level, there is a minimal (<10 m$^3$/m) difference in the amount of dune toe retreat mitigated by the fenced dune between the longest and shortest storms (Figure 12D, E, F). Additionally, the aspect ratio of the dune behind the fence plays a minimal role in influencing volume loss (because the natural dune is never impacted until the fenced dune is eroded) except in the case of the most intense storms when the lowest aspect ratio dunes performs better than the higher aspect ratio dunes (Figure 12C).
* * *
**Section Comparison with Field Surveys:**
**Line 397: "…a weak relationship…"**
**I cannot see this relation, not even weak. But you could plot the aspect ratio vs the volume change in order to make your point clearer. I see high aspect ratio are the ones showing smaller vol change and lower aspect with the greatest vol change. Is this actually in agreement with what you showed…not very clear.**

We have revised these lines for clarity to demonstrate the comparison between the field data and the results from the toes-aligned simulations.

Original: The field data show a weak relationship between dune aspect ratio and erosion (sand volume loss). However, similar to model results for the toe-aligned (constant beach width) dune configurations, those profiles with a lower aspect ratio dune experience similar or even less erosion than high aspect ratio dunes with the same beach width (i.e., at a beach width of 40 m in Fig. 13).

Revised sentence: Similar to model results for the toe-aligned (constant beach width) dune configurations, comparing volume loss for dunes fronted by equally wide beaches, profiles with a lower aspect ratio dune generally experienced similar or even less erosion than high aspect ratio dunes with the same beach width. For example, the field profiles with a beach width of ~30 m show slightly more erosion for the higher aspect ratio dunes compared to the lower aspect ratio dunes (Figure 13).
* * *
**Lines 399-400: "those profiles with a lower aspect ratio dune experience similar or even less erosion than high aspect ratio dunes with the same beach width (i.e., at a beach width of 40 m in 400 Fig. 13)".**
**Once again, this is not clear to me, the diff for the 40m is that you have only one point with low aspect ratio, but the high aspect ratio shows a large variability for the 40m, so this is not clear at all.**

Considering the points clustered around the 40 m beach width, the lowest aspect ratio dune increased in volume while the high aspect ratio dunes all eroded (with variability). The profiles clustered around 30 m beach widths can also be considered here which show slightly more erosion for the highest aspect ratio dunes compared to the low aspect ratio dunes. We have adjusted this in the text to better highlight the result from the field data.

Original sentence: Similar to model results for the toe-aligned (constant beach width) dune configurations, comparing volume loss for dunes fronted by equally wide beaches, profiles with a lower aspect ratio dune generally experienced similar or even less erosion than high aspect ratio dunes with the same beach width (i.e., at a beach width of 40 m in Fig. 13).

Revised sentence: Similar to model results for the toe-aligned (constant beach width) dune configurations, comparing volume loss for dunes fronted by equally wide beaches, profiles with a lower aspect ratio dune generally experienced similar or even less erosion than high aspect ratio dunes with the same beach width. For example, the field profiles with a beach width of ~30 m show slightly more erosion for the higher aspect ratio dunes compared to the lower aspect ratio dunes (Figure 13).
* * *
**Discussion section:**
**Line 435: "accretion occurring at the dune toe for the high aspect ratio dunes that weren't…"**
**Accretion at the toe, but dune crest retreat, right?**

We don't quantify dune crest retreat in this paper, but if the dune is scarping/avalanching then the crest position will retreat once the dune is eroded back to and beyond the initial crest position. We have clarified this in the revised manuscript with a parenthetical comment as shown below.

Original sentence: The high aspect ratio dunes are more likely to collapse when scarped because of avalanching as the dune face slope approaches an angle of repose. This process also likely

explains accretion occurring at the dune toe for the high aspect ratio dunes that weren't completely eroded (Palmsten and Splinter, 2016).

Revised sentence: High aspect ratio dunes are more likely to collapse when scarped because avalanching is likely to occur as the dune face slope approaches the angle of repose. Avalanching also likely explains accretion occurring at the dune toe for the high aspect ratio dunes (Palmsten and Splinter, 2016; *which may also be accompanied by dune crest retreat as the dune face is eroded beyond the initial crest position*).
* * *
**Line 445: "resilient"**
**Do you mean resistant?**

We changed "resilient" to "resistant here
* * *
**Line 448-449: "While dune morphology plays a primary role in describing how dunes erode, particularly with respect to whether or not sediment is piled at the toe of the dune (high aspect ratio dunes) or transported offshore"**
**Why particularly? From your results and interpretation, I can only see with respect to this, so if the aspect is important with respect to additional aspects it should be clearly stated.**

We have revised the sentence in question to remove the qualifier "particularly.": While dune morphology plays a primary role in describing how dunes erode, with respect to whether sediment is piled at the toe of the dune (high aspect ratio dunes) or transported offshore (low aspect ratio dunes), it plays a secondary role to the beach morphology in terms of explaining the amount of erosion that will occur.
* * *
**Line 453: "Wider beaches lead to less sediment loss from the dune and more progradation of the dune toe."**
**This has not been explained, which process is this? Is only avalanching? If so, how do you explain that low aspect ratio dunes also prograde???**

We have revised this sentence for increased clarity: Wider beaches lead to less sediment loss from the dune and *a* more *seaward post-storm location* of the dune toe.

This is also now consistent with the revised explanation in results section 3.2 (see also the explanation at the bottom of Page 8 in this response).
* * *
**Line 457-458: " beach width is the primary control on dune erosion, followed by dune width, and then dune height."**
**I agree with the first part of the sentence, but i do not agree with the affirmation that dune width controls dune erosion, it may control the relative erosion, meaning that if your dune is narrow, it will disappear faster, but the width does not influence the process of erosion itself as does the beach width, which determines the wave energy impacting the dune. alternatively, i agree that high aspect may help erode the dune.**

We have rewritten this sentence for increased clarity:

This result, combined with the results of the toe-aligned scenarios confirms that beach width is the primary control on the volume of sediment eroded from dunes during storms. Our results further suggest that dune width and dune height also affect the volume of sediment eroded by determining the "longevity" of the dune (thereby affecting the transition from avalanching to overwashing, e.g.) under erosive conditions.
* * *
**Lines 469-471: "Thus, the aspect ratio of the natural dune is secondary to the morphology of the fenced dune in providing protection to back-barrier environments (a taller fenced dune would offer even greater protection)."**
**I think that here there are mixing concepts, i think that if you would address this as volumes of sand as defence, distributed within the fenced dune and the additional beach width because of the space needed to place an artificial dune, it would make much more sense to me.**

Our simulations with the fenced dune do not consider variations in the beach width (see Page 9 of this response) and in fact control for the beach width. The decrease (and differences) in erosion observed in the natural dunes in these simulations is due to the fenced dune serving as a barrier between the runup and the natural dune. The role played by the "morphology of the fenced dune" is explained in the preceding sentence, which has been revised for clarity:

We find that the small dune formed by fencing can significantly decrease dune erosion by providing a barrier that must be removed by erosion before the "natural" dune behind it is impacted. In our simulations, the fenced dune was not sufficiently eroded until ~60 hours into the storm, which prevented the dune behind it from experiencing the peak of the storm (Figure 4). The key dynamic in this case (regardless of actual storm duration), was that the fenced dune was sufficiently high *and wide* to protect the natural dune until the peak of the storm had passed. Thus, the aspect ratio of the natural dune is secondary to the morphology of the fenced dune in providing protection to back-barrier environments (a taller fenced dune would offer even greater protection).
* * *
**Line 471: "can reform"**
**Do you actually mean reform? From their work I understood form or develop.**

Given that the paper being referred to (Charbonneau and Wnek, 2016) discusses dune recovery following a storm, we used the word "reform." However, we have changed it in the revised manuscript to "form" following the reviewer's comment.
* * *
**Line 472: "can sand fences effectively prevent storm-induced erosion,"**
**Sand fences do not prevent storm erosion, but the volume of sand accumulated or the sand fencing or the fenced dune.**

Revised sentence:
Charbonneau and Wnek (2016) demonstrated that fenced dunes (*especially when paired with nourishment*) can form quickly (on the order of months) *meaning that not only can the fenced*

*dunes effectively prevent storm-induced erosion, but it is possible for them to recover prior* to the next storm if the frequency of storm impacts is sufficiently low, and assuming the fences are still present following the storm or are re-built.
* * *
**Line 476: "are still present following the storm or are re-built."**
**In their work, they also mention nourishment in those areas, so is not only the fences, but the extra sand that helps, don´t you agree?**

We agree that nourishing the beach helps facilitate fenced dune growth (Itzkin et al., 2020) and that nourishing the beach will limit erosion (Lines 374-387) but the phrase highlighted here by the reviewer is referring to construction of the physical fence and their condition following a storm. We have modified the sentence for clarity in the revised manuscript:

Charbonneau and Wnek (2016) demonstrated that fenced dunes (*especially when paired with a nourishment*) can form quickly (on the order of months) *meaning that not only can the fenced dunes effectively prevent storm-induced erosion, but it is possible for them to recover prior* to the next storm if the frequency of storm impacts is sufficiently low, and assuming the fences are still present following the storm or are re-built.
* * *
**Lines 478-479: "widens the dune but does not add to its elevation will cause the dune to assume a lower aspect ratio than it had in its pre-management state."**
**Of course, and I insist, this implies widening the beach as you state below, and in fact i would change the order and start stating that any management practice aiming to enlarge the dune needs to be paired with nourishment.**

This comment by the reviewer is referring to the following set of sentences:

Any management strategy that widens the dune but does not add to its elevation will cause the dune to assume a lower aspect ratio than it had in its pre-management state. Further, if management is not paired with a beach nourishment (i.e., Itzkin et al., 2020) then the wider dune will likely come at the cost of a slightly narrower beach. The lower aspect ratio (Figure 8) could serve to reduce erosion, but this potential decrease in erosion may well be offset by increased erosion associated with the narrower beach (Figure 10).

While we agree that strictly following the results of this study, dune management should be paired with a nourishment, we are being intentional with the wording throughout the paper to avoid unintentionally advocating for increased instances of nourishment. The suggested revision here could do that and thus it would be our preference to avoid saying that nourishment is needed.
* * *
**Lines 491-492: "slope ($\beta f$), which lowers incident band swash (e.g., Ruggiero et al., 2004) and total wave runup (Stockdon)"**
**You could explain this better and make sure you explain what slope you are refereeing to. Usually, these works use the foreshore slope, and the foreshore+nearshore, i understand that the beach width increase in your case is related to the backshore width, should not you compute the slope of the intertidal beach? how is the backshore included in the equation**

**from stockdon? the slope of the beach should be constant if you change the position of the toe of the dune with not additional changes in the morphology of the beach.**

The reviewer here is referring to the following sentence:

For a given dune aspect ratio and wave duration and intensity, the only difference between the simulations is the increase in beach width (toe-aligned < heel-aligned). This increase in beach width decreases beach slope ($\beta_f$), which lowers incident band swash (e.g., Ruggiero et al., 2004) and total wave runup (Stockdon et al., 2006), reducing the likelihood of dune erosion.

The beach slope being used in this manuscript is consistent with the beach slope used by Stockdon et al. (2006), which is measured from MHW to the dune toe. We also explain this in the methods section (Lines 106 -108), which we have further modified for clarity:

Beach width is calculated as the cross-shore distance between MHW and $D_{low}$ at every time step. Given that pre-storm $D_{low}$ was held constant across all simulations, the beach slope *(measured between MHW and $D_{low}$)* is inversely proportional to the beach width in our simulations (i.e., beach slope decreases as beach width increases).
* * *
**Line 492: "reducing the likelihood of dune erosion."**
**Not because of the slope, but because of the beach width of the dry beach. If your beach is widening because you move the dune behind in your synthetic profiles, it means that what actually grows is the width of the beach berm, which implies large volume of sediment but that does not affect the wave runup.**

The reviewer's comment refers to the following sentence:

For a given dune aspect ratio and wave duration and intensity, the only difference between the simulations is the increase in beach width (toe-aligned < heel-aligned). This increase in beach width decreases beach slope ($\beta_f$), which lowers incident band swash (e.g., Ruggiero et al., 2004) and total wave runup (Stockdon et al., 2006), reducing the likelihood of dune erosion.

In this instance, it is fair to say that the slope is reducing the likelihood of erosion because the slope is being measured between MHW and the dune toe. Ruggiero et al. (2004) and Stockdon et al. (2006) demonstrate how this decrease in the slope limits wave runup and wave energy reaching the dune.
* * *
**Line 508: "thereby increasing the likelihood of eventual barrier drowning"**
**I still don' t get this point. If you are adding sand to your beach/dune, you may not have rollover because the system is able to cope with sea-level rise and maintained, why should it drown? The nourishment can even help build the backbarrier through aeolian transport. I do agree and I am glad to read that the main problem is the sustainability of these type of measures.**

The reviewer's comment refers to the following sentences:

It is important to recognize that although management initiatives such as widening beaches and building dunes with particular aspect ratios can be effective at mitigating erosion during a single storm, these actions may have effects that are undesirable in the long-term as the effects of multiple storms compound. For example, overwash facilitates barrier rollover—a process that is necessary if islands are to maintain subaerial exposure *in the long-term future* as sea level rises (e.g., Leatherman, 1979; Moore et al., 2010; Lorenzo-Trueba & Ashton, 2014; Rogers et al., 2015). Thus, constructing dune and beach systems that reduce the amount of overwash that would otherwise naturally occur may inhibit rollover, thereby increasing the likelihood of eventual barrier drowning (e.g., Magliocca et al., 2011; Rogers et al., 2015).

The barrier island literature demonstrates that developed barrier islands are susceptible to drowning because the flux of sediment to the bay side of the island is limited or even prevented (i.e., Magliocca et al., 2011; Rogers et al., 2015) and even with nourishment on the frontside, drowning can occur from the backside as sea level rises in the more distant future. This is what we are pointing to here. We have added the italicized text above because, yes, for some time, nourishment efforts may offset sea level rise.

**Conclusions section:**
**Line 518: "dunes, although high aspect ratio dunes offer greater protection against more intense storms."**
**Can you expand this? If i got it right, it is not the case, from the results section as you do not properly explain the case of the overwashed dunes. The volume of erosion is of the same order of magnitude for the high ratio dunes (Fig. 8).**

This comment is referring to the following sentence:
We find that low aspect ratio (lower and wider) dunes are more resistant to erosion from increased storm duration than high aspect ratio (taller and narrower) dunes, although high aspect ratio dunes offer greater protection against more intense storms.

It's not clear how we do not properly explain overwash here, as we do not discuss (or observe) overwash in any of the simulations presented in this iteration of the manuscript. The differences in erosion between the low- and high-aspect ratio dunes as a function of storm duration are discussed in the results section 3.1.1.

**Lines 520-522: "eroded sediment is lost offshore whereas the high aspect ratio dunes lose greater amounts of sediment through persistent scarping, more of the sediment is preserved at the toe of the dune as a result of avalanching."**
**You mention this here, but how did you see this? it cannot be easily taken from the results.**

This comment refers to the following sentence, which we have revised for clarity:
For low aspect ratio dunes, eroded sediment is lost offshore whereas the high aspect ratio dunes lose greater amounts of sediment through persistent scarping, more of the sediment is preserved at the toe of the dune, *consistent with the piling up of sediment at the base of the dune as would occur* as a result of avalanching.

In results section 3.1.1 we show that the dune toe location tends to move seawards for the high aspect ratio dunes while it tends to move landwards for the low aspect ratio dunes despite all dunes experiencing a loss in volume. Overwash does not occur so the sediment must be moving seaward from the dunes. The seaward movement of the dune toe for high aspect ratio dunes is consistent with sediment piling up at the base of the dune from avalanching (we explain that the high aspect ratio dunes are close to the angle of repose), but since we cannot observe this directly in the model we have added the italicized text above to clarify the consistency with avalanching
* * *
**Lines 528-529: the sentence is missing something.**

It is not clear which sentence is being referred to here by the reviewer as the line numbers do not correspond to the paper but we did revise the following sentence which was incomplete:

Our results indicate that a tall, wide foredune fronted by a fenced dune and a wide beach *offer the greatest protection from erosion*.
* * *
**Lines 544-545: "management initiatives reduce overwash flux, which is essential for barrier islands to maintain elevation as sea level continues to rise"**
**Not only because of this, but also because of sustainability of the resources needed. You may also need to keep in mind that if you nourish a beach, what also may happen is that the dune will rise and likely the backbarrier because of aeolian sed transport, which is not mentioned here but also contributes to keep the elevation of the coastal barrier.**

Please see our response to the comment at the top of Page 15 of this document, which also refers to this concept. We do state in the paper here that there is a sustainability issue:

Given the challenges of achieving such a foredune morphology in the face of rising sea level and within resource limitations (i.e., sand availability, cost, etc.), our findings suggest that the greatest increase in short-term protective service can be achieved by widening beaches, regardless of the frontal dune morphology.

Our comment on the need for overwash for barrier survival is in line with the literature that looks at the survivability of developed barrier islands over long timescales (i.e., Magliocca et al., 2011, Rogers et al., 2015). We do refer to managed retreat, which suggests a way to widen a beach without advocating for nourishment (see the bottom of Page 13 of this document):

Alternative strategies for widening beaches would also have a similar protective effect (e.g., managed retreat; Cutler et al., 2020; Gibbs, 2016) while allowing for more of the natural processes to occur that allow an island to evolve and persist in the face of rising sea levels. Along the NC coast, aeolian sediment transport to the back-barrier is minimal (and removed from roads and properties, for example) because the dunes are well vegetated. We are not aware of studies that indicate aeolian sediment transport is sufficient to raise island elevation to avoid drowning from the backside, but this is an interesting concept that would be fun to explore further.